# A widely-occurring family of pore-forming effectors broadens the impact of the *Serratia* Type VI secretion system

Mark Reglinski[1], Quenton W Hurst [ID][2], David J Williams [ID][1], Marek Gierlinski[1], Alp Tegin Şahin [ID][1], Katharine Mathers[1], Adam Ostrowski[1], Megan Bergkessel [ID][1], Ulrich Zachariae [ID][1], Samantha J Pitt [ID][2] & Sarah J Coulthurst [ID][1]✉

## Abstract

**Delivery of antibacterial effector proteins into competitor cells using the Type VI secretion system (T6SS) is a widespread strategy for inter-bacterial competition. While many enzymatic T6SS effectors have been described, relatively few which form pores in target cell membranes have been reported. Here, we describe a widely-occurring family of T6SS-dependent pore-forming effectors, exemplified by Ssp4 of *Serratia marcescens* Db10. We show in vitro that Ssp4 forms regulated pores with high selectivity for cations, and use structural models and molecular dynamics simulations to predict how these pores conduct ions. Ssp4 has a broader phylogenetic distribution and is active against a wider range of bacterial species than Ssp6, the other pore-forming effector delivered by the same T6SS, with the two effectors displaying distinct ion selectivities and impacts on intoxicated cells. Finally, identification of Ssp4-resistant mutants revealed that a *mucA* mutant of *Pseudomonas fluorescens* is protected against T6SS attacks. We propose that deployment of two distinct T6SS-dependent pore-forming toxins is a common strategy to ensure effective de-energisation of closely- and distantly-related competitors.**

**Keywords** Type VI Secretion System; T6SS; Pore-forming Effector; Antibacterial Toxin; Bacterial Protein Toxin
**Subject Category** Microbiology, Virology & Host Pathogen Interaction

## Introduction

Bacteria typically exist in mixed microbial communities, where competition for resources between and within species represents a constant challenge and a key driver for the evolution of antimicrobial toxins and the molecular machineries that deliver them to competitor cells (Granato et al, 2019; Peterson et al, 2020). The Type VI secretion system (T6SS)

occurs widely in Gram-negative bacteria and is used to deliver anti-bacterial toxins, or effector proteins, directly into neighbouring competitors, effectively killing or disabling recipient cells (Coulthurst, 2019). Growing evidence supports a role for the T6SS in determining the success of individual isolates and the composition of the polymicrobial population in a wide variety of host-associated and environmental communities (Allsopp et al, 2020; Gallegos-Monterrosa and Coulthurst, 2021; Robitaille et al, 2023). The T6SS is a dynamic protein nanomachine in which contraction of an intracellular sheath structure anchored on a membrane-bound basal complex propels a cell-puncturing device out of the secreting cell and into a neighbouring recipient, or target, cell. This expelled puncturing structure, comprising a tube of Hcp tipped with a spike of VgrG and PAAR proteins, is decorated with multiple effector proteins, which either interact with or are covalently fused to the tube or spike proteins, thereby delivering the effectors into the target cell (Jurenas and Journet, 2021; Wang et al, 2019).

Since the discovery of inter-bacterial competition mediated by the T6SS (Hood et al, 2010), an ever-increasing number of T6SS-delivered anti-bacterial effectors have been reported, with molecular targets that are conserved and essential in bacterial cells (Jurenas and Journet, 2021). The vast majority of these are enzymatic toxins, including families of effectors which cleave the peptidoglycan cell wall, hydrolyse nucleic acids, degrade membrane phospholipids, deplete NAD or ATP, and modify RNA or proteins by ADP-ribosylation (Bullen et al, 2022; Hernandez et al, 2020; Jurenas and Journet, 2021). Self-intoxication and intoxication by incoming effectors delivered by genetically identical neighbouring cells is prevented through the co-expression of effector-specific immunity proteins. Immunity proteins are localised at the site of action of their cognate effector, bind tightly and specifically to the effector, and, for enzymatic toxins, typically inhibit effector activity by blocking the active site (Hernandez et al, 2020). The mode of action of enzymatic effectors can often be predicted by sequence homology or structural prediction, including, more recently, with the use of AlphaFold2, which can typically generate reliable predictions for soluble enzymatic domains (Jumper et al, 2021). A small number of T6SS effectors have been reported, which, by contrast, act as pore-forming toxins in recipient cell membranes,

[1]School of Life Sciences, University of Dundee, Dundee DD1 5EH, UK. [2]School of Medicine, University of St Andrews, St Andrews KY16 9TF, UK.
✉E-mail: s.j.coulthurst@dundee.ac.uk

including two shown to form cation-selective pores in vitro (Gonzalez-Magana et al, 2022; Mariano et al, 2019). However, identification of such effectors has been hampered by the fact that their function is much harder to predict in silico, given that they typically present as small proteins with no homologues of known structure or function. We believe that pore-forming effectors represent a larger contribution to the pool of anti-bacterial effectors than is currently appreciated.

*Serratia marcescens* is an opportunistic pathogen which is widespread in diverse environmental niches and represents a significant cause of antibiotic-resistant hospital-acquired infections (Mahlen, 2011). The T6SS of the model strain *S. marcescens* Db10 is well characterised and displays potent anti-bacterial and anti-fungal activity. Ten effectors delivered by this T6SS have been identified by secretomic and genetic studies, including anti-bacterial effectors with peptidoglycan amidase (Ssp1 and Ssp2), DNase (Rhs2) and NAD(P)$^+$-glycohydrolase (Rhs1) activity, and two anti-fungal effectors (Cianfanelli et al, 2016; Fritsch et al, 2013; Hagan et al, 2023; Srikannathasan et al, 2013; Trunk et al, 2018). This T6SS also delivers the ion-selective pore-forming effector Ssp6, which forms cation-selective pores in vitro and, as a consequence, disrupts the inner membrane potential in vivo (Mariano et al, 2019). Another effector, Ssp4, was identified in the T6SS-dependent secretome of *S. marcescens* Db10 and subsequently shown to possess anti-bacterial activity, which was observable upon expression in the periplasm of *E. coli* and neutralised by a cognate immunity protein, Sip4 (Fritsch et al, 2013). However, no function could be readily ascribed to Ssp4, given a lack of sequence or structural similarity with any T6SS-associated or other proteins described previously.

Here, we report that Ssp4 forms ion-selective pores in the membrane of susceptible bacterial cells and represents the founding member of a widely occurring new family of T6SS-dependent pore-forming toxins. Importantly, we show that Ssp4 is active against a wider range of target species than the other cation-selective pore-forming toxin delivered by this T6SS, Ssp6, revealing that not all T6SS anti-bacterial effectors have broad-spectrum activity. We further demonstrate that the ion selectivities of the pores formed by Ssp4 and Ssp6 are distinct and provide a first high-resolution model of a membrane pore formed by a T6SS-delivered effector. Our data support a model whereby possession of two distinct T6SS-dependent pore-forming toxins ensures effective de-energisation of both closely- and distantly related competitors.

## Results

### The properties of Ssp4 and Sip4 are consistent with Ssp4 being a membrane-targeting effector

Ssp4 is encoded with its cognate immunity protein, Sip4, within a three-gene insertion in the isoleucine, leucine and valine (*ilv*) gene cluster of *S. marcescens* Db10 (Fig. 1A). AlphaFold2 (Jumper et al, 2021) was used to generate a high-confidence structural prediction for Ssp4, comprising 13 α-helices and one β-sheet (pLDDT value 85.9, Fig. 1B,C). Regions within three of these α-helices were predicted to represent transmembrane helices by MEMSAT (Fig. 1B,C), indicating that Ssp4 may have the ability to integrate into membranes. However, no Ssp4 homologues of known

or predicted function were identified in sequence databases, and no convincing structural homologues were retrieved when the predicted structure of Ssp4 was used to search the Protein Data Bank.

In order to investigate the mode of toxicity of Ssp4, *E. coli* MG1655 was transformed with plasmids directing the expression of Ssp4, either retained in the cytoplasm or fused to an N-terminal signal peptide for export to the periplasm (sp-Ssp4). Inducing expression of sp-Ssp4 inhibited further growth but did not cause a drop in optical density, suggesting that Ssp4 does not cause lysis of intoxicated cells (Fig. 1D). No inhibition was observed when Ssp4 was retained in the cytoplasm or when Sip4 was co-expressed. Next, we determined whether the action of Ssp4 results in growth inhibition at a single cell level when it is delivered into neighbouring cells by the T6SS. Time-lapse fluorescence microscopy was used to observe co-cultures between cells of wild type, T6SS-inactive (Δ*tssE*), or Δ*ssp4* strains of *S. marcescens* Db10 expressing mCherry ('attacker') and an Ssp4-susceptible 'target' strain, Db10 Δ*ssp4*Δ*sip4* expressing GFP (this mutant lacks the protection normally conferred by Sip4). Target cells in contact with wild-type attacker cells generally failed to proliferate and divide, whilst isolated target cells or target cells in contact with T6SS-deficient or Ssp4-lacking attackers proliferated indistinguishably from attacker cells (Fig. 1E). Given also that no target cell lysis events were observed, our combined data are consistent with Ssp4 intoxication causing growth inhibition but not cell lysis.

The subcellular localisation of T6SS immunity proteins typically indicates the site of action of the cognate effector (Coulthurst, 2019). Using MEMSAT, Sip4 was predicted to contain four transmembrane helices, with an N- and C-terminal 'in' topology (Fig. 1F). This prediction was consistent with a low confidence AlphaFold model in which Sip4 folds into a bundle of seven α-helices (pLDDT 64.9; Appendix Fig. S1) and suggested that Sip4 is an integral membrane protein. To confirm the membrane insertion and topology of Sip4, fully functional variants of Sip4 with no cysteine residues or selected single cysteine substitutions were expressed in the Db10 Δ*ssp4*Δ*sip4* mutant (Fig. 1F; Appendix Fig. S2), and cells were incubated with the membrane impermeant, Cys-reactive reagent mPEG-malemide (mPEG-Mal). Sip4 variants with cysteine substitutions in residues predicted to be located in the cytoplasm (S18C and G43C) or a transmembrane helix (G145C) were only labelled with mPEG-Mal in the presence of SDS, confirming that these amino acid positions are not exposed extra-cytoplasmically and only become available for labelling upon inner membrane disruption (Fig. 1G). In contrast, the G81C variant could be labelled in the absence of SDS, confirming that Gly 81 is exposed in the periplasm (Fig. 1G). These data confirm that Sip4 is an integral inner membrane protein and are consistent with the predicted topology. Taken together, the predicted α-helical structure of Ssp4, its non-lytic and periplasmic toxicity, and the integral membrane location of its immunity protein, suggested that Ssp4, like Ssp6, was likely to be a membrane-targeting toxin.

### Ssp4 and Ssp6 display distinct target specificities during T6SS-mediated competition

Next, we compared the ability of Ssp4 and the previously-characterised membrane-targeting effector Ssp6 to intoxicate different bacterial species upon delivery by the T6SS. Initially, we compared the T6SS-dependent anti-bacterial activity of Δ*ssp4* and Δ*ssp6* mutants with that of wild type *S. marcescens* Db10 against

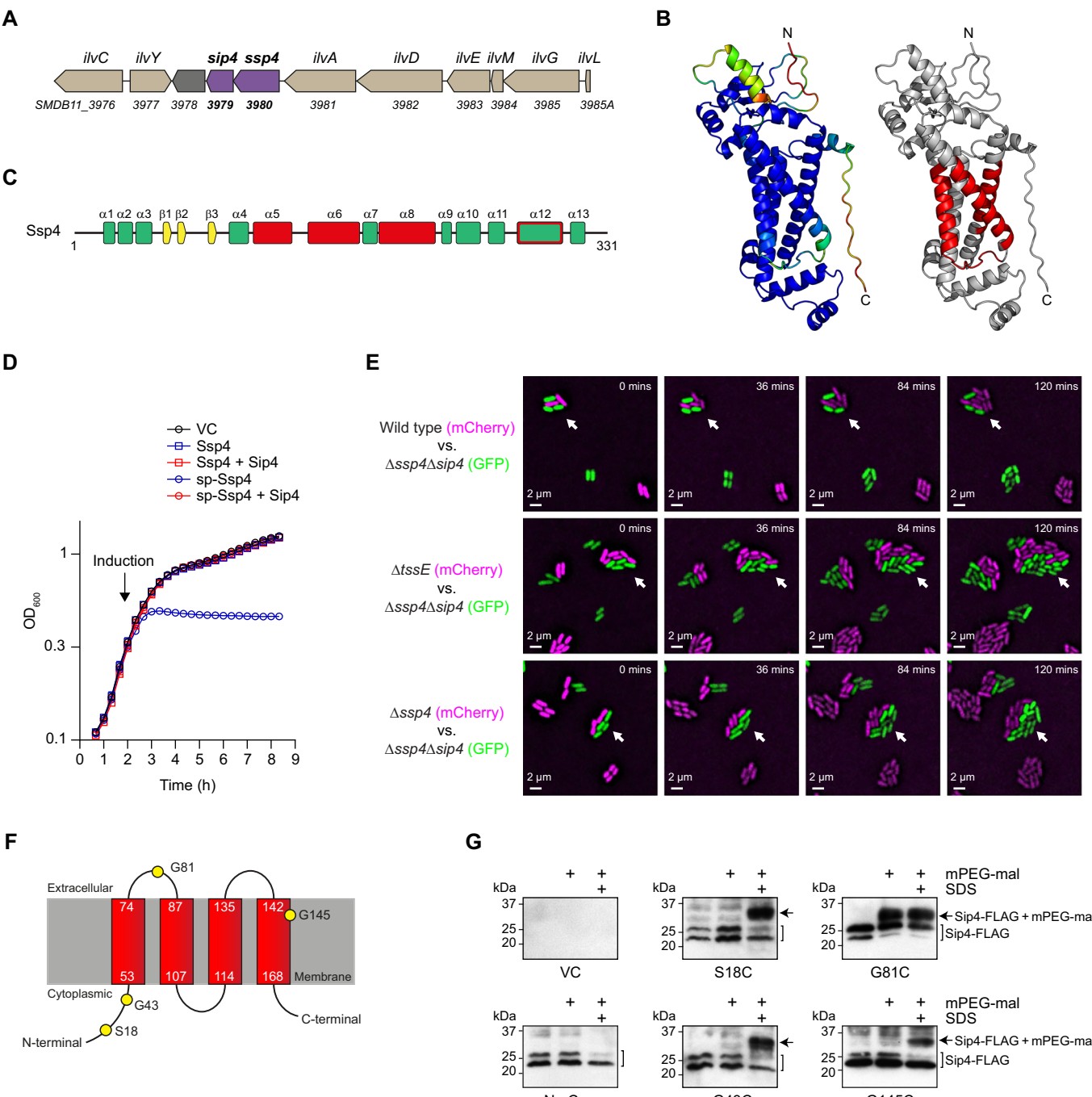

**A** ilvC ilvY sip4 ssp4 ilvA ilvD ilvE ilvM ilvG ilvL
SMDB11_3976 3977 3978 **3979 3980** 3981 3982 3983 3984 3985 3985A

**B**

**C** Ssp4 α1 α2 α3 β1 β2 β3 α4 α5 α6 α7 α8 α9 α10 α11 α12 α13

**D**

**E**
Wild type (mCherry) vs. Δssp4Δsip4 (GFP)

ΔtssE (mCherry) vs. Δssp4Δsip4 (GFP)

Δssp4 (mCherry) vs. Δssp4Δsip4 (GFP)

**F** Extracellular — G81 — G145

Cytoplasmic — G43 — S18 — N-terminal — C-terminal — Membrane

**G**

*Pseudomonas fluorescens* and *E. coli*, by determining the recovery of viable target cells following co-culture with the attacker strain of *S. marcescens*. While the Δssp4 mutant showed a small reduction in activity against *P. fluorescens*, in the other cases, no significant difference with wild-type Db10 was observed (Fig. 2A,B). This is not unexpected, given that there are at least seven other anti-bacterial effectors delivered by the Δssp4 and Δssp6 mutants. Therefore, to observe the impact of Ssp4 and Ssp6 in the absence of the other known anti-bacterial effectors, we separately re-introduced the gene for each effector into *S. marcescens* Db10 lacking all known anti-bacterial effectors (strain Δ9). Against

*E. coli*, both Δ9 + ssp4 and Δ9 + ssp6 displayed considerable anti-bacterial activity compared with Δ9, showing that both effectors can act against this species and that Ssp4 is the more potent (Fig. 2A). Ssp4 also showed substantial activity in *S. marcescens* when Db10 or Δ9 + ssp4 were co-cultured with the Δssp4Δsip4 mutant (Fig. 2C). Unexpectedly, whilst Ssp4 also effectively intoxicated *P. fluorescens*, Ssp6 showed no detectable activity against this target species (Fig. 2B), implying that the two effectors have an overlapping but distinct range of target species. To investigate this observation further, we determined whether T6SS-delivered Ssp4 or Ssp6 was able to intoxicate another member

**Figure 1.  Ssp4 is a non-lytic anti-bacterial toxin that acts from within the periplasm and is neutralised by the integral inner membrane immunity protein Sip4.**

(A) Genomic context of the genes encoding Ssp4 and Sip4 in *S. marcescens* Db10, with genomic identifiers (SMDB11_xxxx) below each gene and gene names above (*ilv*, isoleucine, leucine and valine genes). (B) Structure of Ssp4 predicted by AlphaFold2. Left, structure is coloured by pLDDT value, spectrum from red (<50) to blue (>90). Right, regions predicted to form transmembrane helices by MEMSAT2 are highlighted in red. (C) Secondary structure elements in the predicted structure of Ssp4, with α-helical regions predicted to include transmembrane helices by MEMSAT coloured red; the predicted structure of Ssp4 indicates that helix α12 should also cross the membrane, as shown by a red outline. (D) Growth of *E. coli* MG1655 carrying the vector control (VC, pBAD18-Kn) or plasmids directing the expression of Ssp4 or Ssp4 fused with an N-terminal OmpA signal peptide (sp-Ssp4), either alone or with Sip4. Gene expression was induced by the addition of 0.2% ʟ-arabinose at the time indicated. Points show mean ± SEM (*n* = 3 biological replicates). (E) Ssp4-mediated growth inhibition observed by time-lapse fluorescence microscopy. An Ssp4-susceptible target, *S. marcescens* Δ*ssp4*Δ*sip4* expressing cytoplasmic GFP (green), was co-cultured with wild type (WT) or mutant (Δ*tssE* or Δ*ssp4*) attacker strains expressing cytoplasmic mCherry (magenta) for 2 h. Arrows highlight example microcolonies where attacker and target cells are in contact. Scale bar 2 μm. Images are representative of three independent experiments, including at least nine frames per attacker strain. (F) Predicted membrane topology of Sip4 generated by MEMSAT with amino acids selected for cysteine substitution represented as yellow circles. (G) Cells of *S. marcescens* Δ*ssp4*Δ*sip4* carrying the vector control (VC, pSUPROM) or plasmids directing the expression of Sip4-FLAG with native Cys residues mutated to Ala (No Cys) or derivatives carrying the Cys substitutions indicated, were treated with mPEG-MAL in the presence or absence of SDS. Sip4-FLAG and Sip4-FLAG-mPEG-MAL species were detected by immunoblotting. Source data are available online for this figure.

of the Enterobacterales, *Enterobacter cloacae*, and another more-distantly related species, *Burkholderia thailandensis*. Again, Ssp4 was active against both species, whilst Ssp6 was only active against *Ent. cloacae* (Fig. 2D,E), confirming the distinct target ranges of the two effectors. The observation of a modest decrease in target cell recovery in the presence of the Δ9 strain compared with the T6SS-inactive Δ*tssE* mutant for several of the target species suggests that at least one more, yet-to-be-identified T6SS effector exists in *S. marcescens* Db10. It is possible that synergy with this unknown effector(s) may be contributing to the Ssp4- and Ssp6-dependent anti-bacterial activity displayed by Δ9 + *ssp4* and Δ9 + *ssp6* against non-*Serratia* target species.

## Ssp4 causes loss of membrane potential in susceptible cells

Given the phenotypic similarity of Ssp4 with other membrane-targeting effectors, we investigated whether Ssp4 affects the membrane potential or permeability of intoxicated cells. We co-cultured attacker strains delivering Ssp4 (wild type Db10 or Δ9 + *ssp4*) with the Ssp4-susceptible target Δ*ssp4*Δ*sip4*, and stained the total mixed population with the voltage-sensitive dye DiBAC$_4$(3) and propidium iodide (PI). Around 15–20% of the total population (attacker and target cells) were positive for DiBAC$_4$(3) fluorescence, indicating that the cells had become depolarised. However, these cells did not simultaneously stain with PI, indicating that there was no permeabilisation or loss of membrane integrity (Fig. 3A). The population of DiBAC$_4$(3)-positive, depolarised cells was not observed when the attacking cells were unable to deliver Ssp4 (Δ*tssE*, Δ*ssp4* and Δ9 mutants), confirming that it represents Ssp4-intoxicated cells. Similarly, heterologous expression of sp-Ssp4 in *P. fluorescens* induced membrane depolarisation but not permeabilisation (Fig. 3B). Therefore, Ssp4, like Ssp6 (Mariano et al, 2019), disrupts the inner membrane potential in intoxicated cells, without the formation of large unspecific pores or loss of bilayer integrity. However, no DiBAC$_4$(3) or PI staining above control was observed on expression of sp-Ssp6 in *P. fluorescens* (Fig. 3B), consistent with the lack of Ssp6 activity observed against *P. fluorescens* in co-culture (Fig. 2B).

## Ssp4 forms ion-selective membrane pores in vitro

The observation that Ssp4 intoxication causes inner membrane depolarisation suggested that this protein, like Ssp6 (Mariano et al, 2019), may form ion-selective pores, leading to disruption of the

membrane potential through ion leakage. To test the ability of Ssp4 to form membrane pores, purified monomeric Ssp4 protein (Appendix Fig. S3) was incorporated into artificial lipid bilayers under voltage-clamp conditions. In symmetrical KCl (510 mM KCl in *cis* and *trans* chambers), Ssp4 generated a measurable current when the membrane was clamped at voltages of >40 or <−40 mV, confirming that Ssp4 is capable of ion permeation (Fig. 4A,B). Between 20 and −20 mV, the signal-to-noise ratio was too low to accurately determine current measurements, although pore openings remained visible (Fig. 4A). In symmetrical 510 mM KCl solutions, Ssp4 displayed a single-channel conductance of 18.4 ± 0.64 pS (mean ± SD; *n* = 4), consistent with the conduction of ions through the Ssp4 pore. In order to investigate whether Ssp4 is permeable to cations, anions or both, a current/voltage (*I/V*) relationship in non-symmetrical conditions was determined and used to calculate the reversal potential (*E*$_{rev}$) of the Ssp4 pore. Under these conditions, the Ssp4 pore had a single-channel conductance of 19.4 ± 0.93 pS (mean ± SD; *n* = 4) and the calculated reversal potential was −14.5 ± 1.6 mV (mean ± SD; *n* = 4). This value is closer to the predicted equilibrium potential (calculated using the Nernst equation) of K$^+$ (−22.8 mV) than Cl$^−$ (+22.8 mV), indicating a preference for cations over anions (Fig. 4B). As a control, a sample from a mock purification (in the absence of Ssp4) was added to the *cis* chamber under voltage-clamp conditions an alternating holding command of +80 or –80 mV was applied over a 10 min period. Under these conditions, no currents were observed, confirming that pore formation can be attributed to Ssp4 (Appendix Fig. S4).

In order to establish whether the Ssp4 pore shows a preference for monovalent or divalent cations, the relative permeability of K$^+$ and Ca$^{2+}$ was examined. When Ca$^{2+}$ was the permeant ion in the *cis* chamber and K$^+$ the permeant ion in the *trans* chamber, a Ca$^{2+}$/K$^+$ permeability ratio (*P*Ca$^{2+}$/*P*K$^+$) of 0.46 ± 0.05 was obtained (mean ± SD; *n* = 3), implying that Ssp4 transports monovalent and divalent cations with little discrimination (Fig. 4D). However, when CaCl$_2$ was in the *trans* chamber and KCl in the *cis* chamber, current saturation was apparent at voltages >50 mV, but not at negative potentials, indicative of a change in the behaviour of the pore (Fig. 4C,D). Additionally, current saturation was only observed when high Ca$^{2+}$ was in the *trans* chamber, suggesting that Ssp4 incorporates into the membrane in a fixed orientation. The single-channel conductance of the pore also changed when Ca$^{2+}$ was present in the *trans* chamber, increasing from 15.0 ± 0.40 pS (mean ± SD; Ca$^{2+}$ ions in *cis*) to 17.2 ± 2.8 pS (mean ± SD; Ca$^{2+}$ ions in *trans*), suggesting that the channel may undergo a Ca$^{2+}$-dependent conformational change.

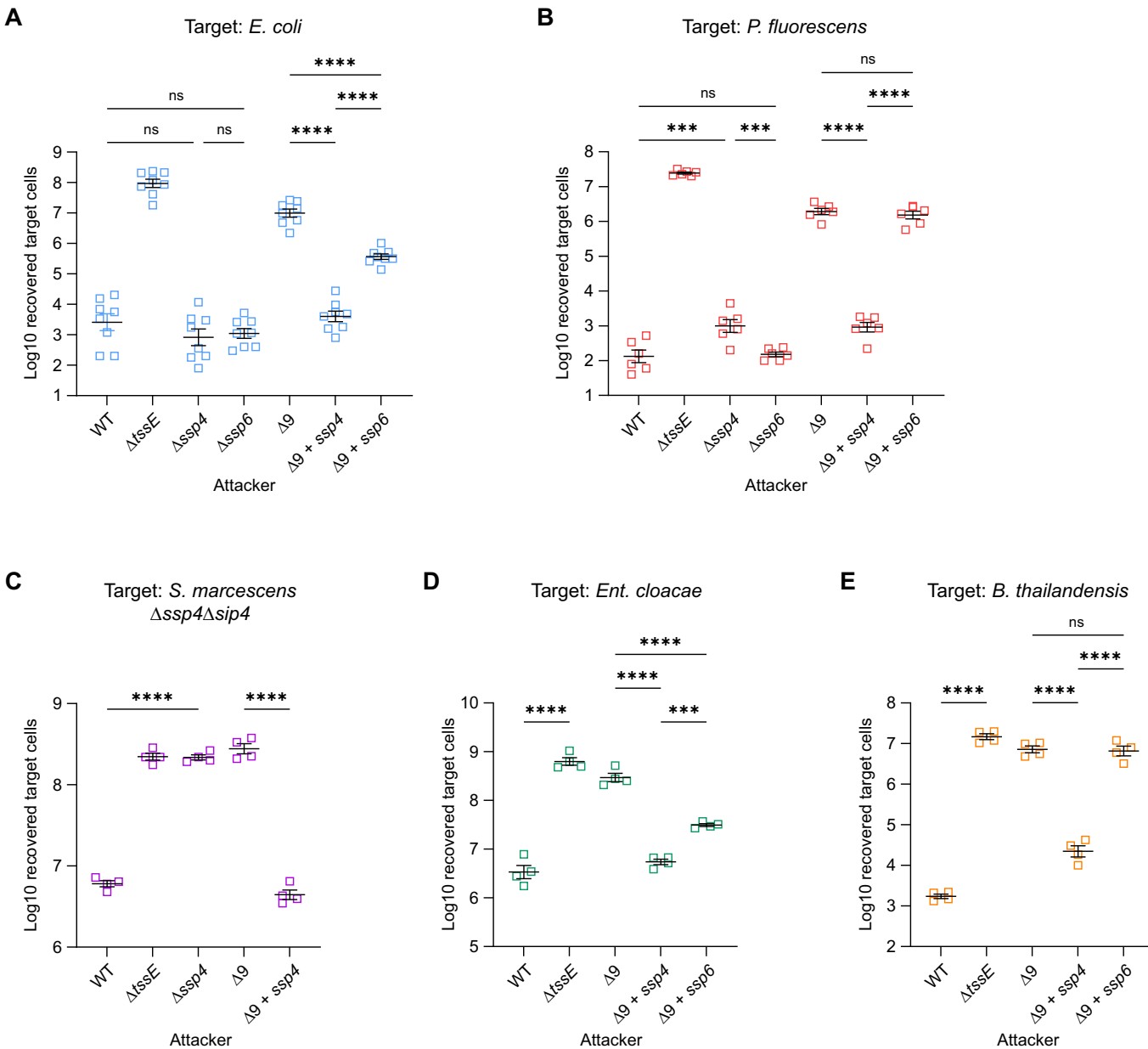

**Figure 2. Ssp4 and Ssp6 differ in their activity against different target species when delivered by the T6SS.**

T6SS-dependent anti-bacterial activity of wild type (WT) or mutant (ΔtssE, Δssp4, Δssp6, Δ9, Δ9 + ssp4 and Δ9 + ssp6) strains of *S. marcescens* Db10, as indicated, against (**A**) *E. coli* BW25112, (**B**) *P. fluorescens* 55, (**C**) *S. marcescens* Db10 Δssp4Δsip4, (**D**) *Ent. cloacae* ATCC13047 or (**E**) *B. thailandensis* E264 target strains. The Δ9 mutant lacks all known anti-bacterial effectors in Db10; Δ9 + ssp4 and Δ9 + ssp6 have the respective effector reintroduced. T6SS-inactive mutants of *Ent. cloacae* and *B. thailandensis* were used to prevent killing of the attacker by the target. Recovery of target cells was enumerated following co-culture of attacker and target at an initial ratio of 1:1 for 4 h. Data were presented as mean ± SEM with individual data points overlaid ($n = 8$, $n = 6$ and $n = 4$ biological replicates for panels (**A–E**, respectively); ****$P < 0.0001$, ***$P < 0.001$, ns not significant, one-way ANOVA with Tukey's test; for clarity, only selected comparisons are displayed. $P$ values from left to right (**A**) $P = 0.5175$, $P = 0.8106$, $P = 0.9990$, $P < 0.0001$, $P < 0.0001$, $P < 0.0001$; (**B**) $P = 0.0004$, $P > 0.9999$, $P = 0.0010$, $P < 0.0001$, $P = 0.9972$, $P < 0.0001$; (**C**) $P < 0.0001$, $P < 0.0001$; (**D**) $P < 0.0001$, $P < 0.0001$, $P < 0.0001$, $P = 0.0001$; (**E**) $P < 0.0001$, $P < 0.0001$, $P = 0.9978$, $P < 0.0001$. Source data are available online for this figure.

## Molecular dynamics simulations of Ssp4 reveal several oligomeric forms capable of function

Having demonstrated the ability of Ssp4 to conduct ions across a membrane, we proceeded to investigate the potential structure and mode of ion conductance of the pores formed by Ssp4 in target cell membranes. We used AlphaFold2 to generate a structural model of a truncated version of the Ssp4 monomer, Ssp4$_{114-331}$, lacking the N-terminal regions, which were disordered in the predicted structure of the full-length protein, and then, based on this truncated version, generated structural predictions for Ssp4 homo-multimers containing different numbers of Ssp4 monomers. The

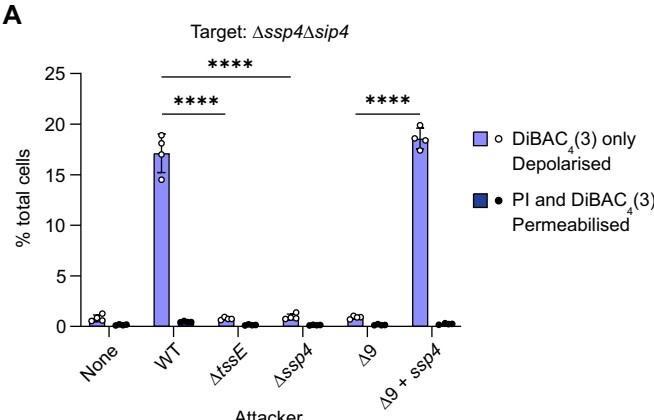

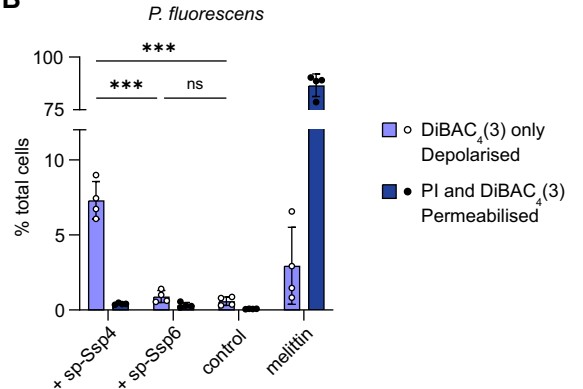

**Figure 3. Intoxication by Ssp4 causes loss of membrane potential.**

(A) The Ssp4-susceptible mutant of *S. marcescens* Db10, $\Delta ssp4\Delta sip4$, was co-cultured with wild type (WT) or mutant ($\Delta tssE$, $\Delta sip4$, $\Delta 9$ and $\Delta 9 + ssp4$) attacker strains, as indicated, at a starting ratio of 1:2 for 4 h, then the mixed population was stained with DiBAC$_4$(3) and propidium iodide (PI) and analysed by flow cytometry to determine membrane potential and permeability, respectively. The percentage of cells in the total bacterial population identified as being depolarised (positive for DiBAC$_4$(3) staining, negative for PI staining), or permeabilised (positive for both DiBAC$_4$(3) and PI staining) quantified. (B) Cells of *P. fluorescens* carrying a chromosomal insertion directing the expression of Ssp4 or Ssp6 fused with an N-terminal signal peptide (sp-Ssp4 or sp-Ssp6) under the control of P$_{Rha}$, or P$_{Rha}$ alone (control), were induced by the addition of 0.05% rhamnose and the percentage of cells depolarised or permeabilised was determined as in (A). Cells treated with melittin acted as a positive control for permeabilisation. Data were presented as mean ± SEM with individual data points overlaid ($n = 4$ biological replicates); ****$P < 0.0001$, ***$P < 0.001$, ns not significant, one-way ANOVA with Tukey's test; for clarity, only selected comparisons are displayed. $P$ values from left to right (A) $P < 0.0001$, $P < 0.0001$, $P < 0.0001$; (B) $P = 0.0002$, $P = 0.0001$, $P = 0.9901$. Source data are available online for this figure.

predicted multimeric structures displayed somewhat lower confidence scores compared with the truncated monomer, but, except for the hexamer, the individual subunits in each multimer showed high structural similarity to the full length Ssp4 monomer (RMSD 0.7–2.9 Å over 164–198 aligned residues) and to the truncated monomer (backbone Cα RMSD values between 2.3 and 3.5 Å over all residues) (Appendix Table S1).

We analysed these potential pore-forming assemblies using molecular dynamics simulations to determine whether they could stably integrate into a lipid bilayer and, if so, whether they formed

hydrated pores to support the conductance of ions. Models based on predicted monomer-octamer oligomeric structures of Ssp4$_{114-331}$ with manual removal of the final 29 amino acids (Ssp4$_{114-302}$) were embedded in model membranes and taken forward for analysis by atomistic molecular dynamics simulations of at least 250 ns length, performed in triplicate for each model. The monomer and dimeric-pentameric complexes showed high structural stability within the membranes, with average simulation backbone RMSD values of 2.2–3.2 Å over 0–250 ns production runs (following equilibration for 100 ns). By contrast, the hexamer and octamer were already structurally unstable during the equilibration phase and the heptamer failed to stably insert into the membrane (Appendix Table S1); the pentamer was also excluded as a potential form of Ssp4 since it generated a large membrane disruption site incompatible with the experimental observations. Based on MEMSAT predictions and the fact that Ssp4 is active from the periplasm (Fig. 1D), suggesting that it enters the membrane from the periplasmic side, we predict that the pore is oriented with its N- and C-termini (including the ~113 amino acid N-terminal domain truncated from the model) in the periplasm.

Having established that monomeric, dimeric, trimeric and tetrameric complexes of Ssp4 could represent relevant biological assemblies based on membrane stability, we examined their ability to conduct ions. We initially focused on the tetrameric assembly due to its channel-like appearance (Fig. 5A). In extended triplicate simulations of 1 μs length at a membrane voltage of 100 mV, we found an average conductance of $15 \pm 6$ pS in 150 mM KCl solution. This value is in good agreement with the experimental conductance of 18.4 pS observed in electrophysiology experiments (Fig. 4). However, strikingly, we noticed that the pore current was generated by the flux of ions across a single subunit at a time (primarily K$^+$ ions, alongside a ~12% contribution of Cl$^-$ ions), and not through a central channel between the Ssp4 subunits (Fig. 5B). Examination of all the stable complexes (monomer-tetramer) revealed that the monomer and each individual subunit in these multimeric complexes readily formed a stable hydrated pore, which persisted during all simulations (Fig. 5C). We consider the hydrated pores in each individual subunit to be generally competent to conduct ions. However, it appears that in the multimeric assemblies, only one subunit actively contributes to ion conduction over each period. Since Ssp4 is predicted to form hydrated pores competent to conduct ions in all of the oligomeric states, which are structurally stable in a membrane environment, we conclude that monomeric, dimeric, trimeric, and tetrameric assemblies could potentially contribute to Ssp4 function. However, since ions are conducted across an α-helical bundle within an individual Ssp4 subunit, oligomerisation is not required to explain pore-forming activity. Given that the monomer displays the highest predicted membrane stability, it is likely to represent the functional unit of Ssp4, at least under certain conditions. Consistent with this, we observed K$^+$ ion permeation in triplicate simulations of monomeric Ssp4 of ~1-μs length at a voltage of ~350 mV (Fig. 5B). After a lag time, in which K$^+$ ions bound tightly to a cluster of polar residues in the centre of the hydrated pore (Appendix Fig. S5), Ssp4 adopted an ion-permeable state with a conductance of $4 \pm 3$ pS. This value is also in a similar range as the observed experimental conductance, particularly as the individual lag time observed may influence the number of permeating ions.

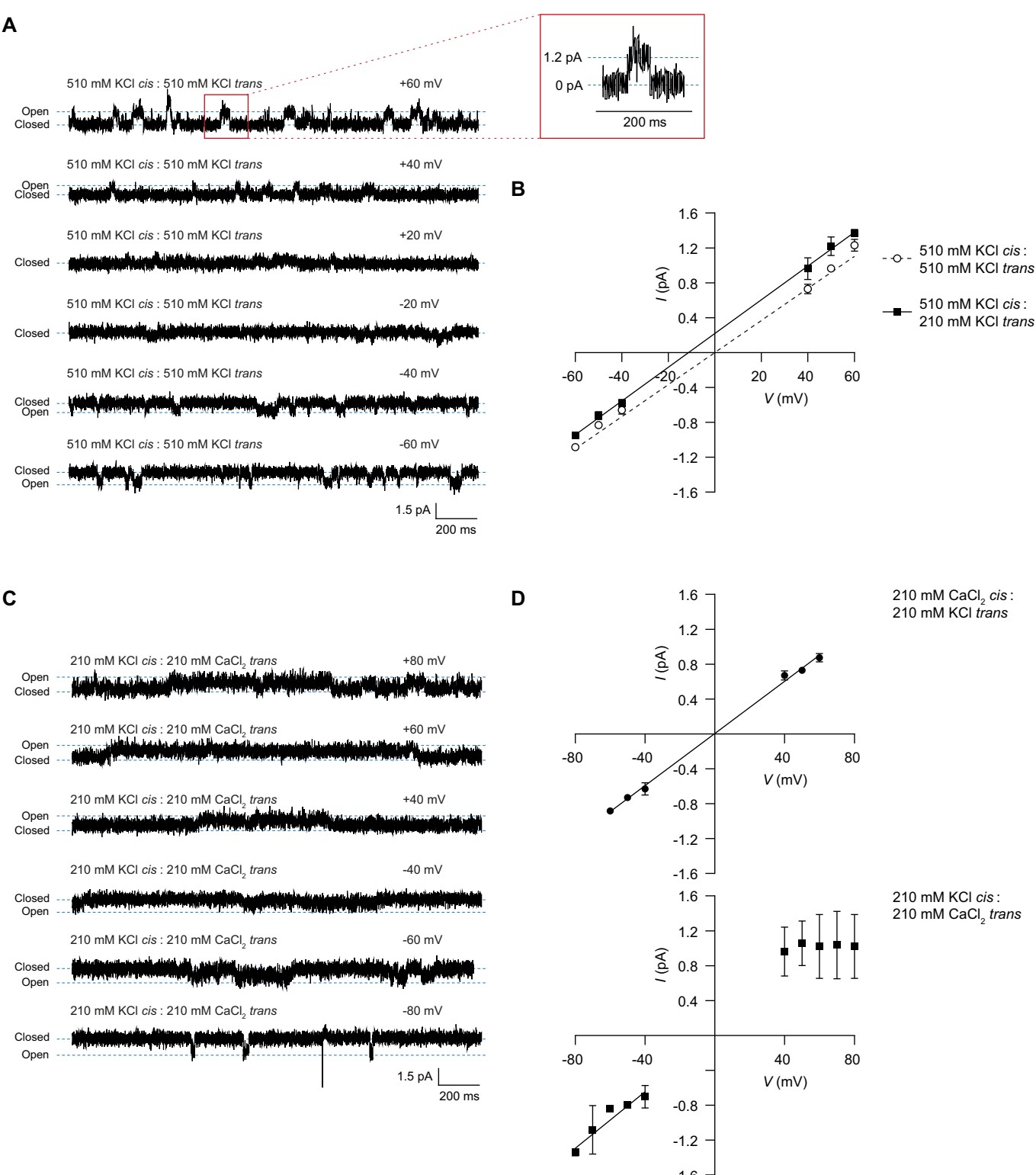

## Ssp4 intoxication is accompanied by increased levels of intracellular reactive oxygen species

Given that Ssp4 and Ssp6 are both ion-selective pore-forming effectors, we next asked if they cause the same response in intoxicated cells. Many antibiotics and other stressors, including phage infection and T6SS attack, have been reported to cause production of reactive oxygen species (ROS) in bacterial cells (Kohanski et al, 2007; Zhao et al, 2015) (Dong et al, 2015; Hong et al, 2019). To determine if ROS generation occurs following Ssp4

**Figure 4. Ssp4 forms ion-selective pores in planar lipid bilayers in vitro.**

Purified Ssp4 was incorporated into artificial lipid bilayers under voltage-clamp conditions, and the resulting current was measured. (A) Representative single-channel experiment in which Ssp4 was added to the *cis* chamber with 510 mM KCl as the source of permeant ions. (B) Current-voltage relationship for Ssp4 with KCl as the source of permeant ions. Dashed line, symmetrical 510 mM KCl; solid line, 510 mM KCl *cis* chamber, 210 mM KCl *trans* chamber. (C) Representative single-channel experiment in which Ssp4 was added to the *cis* chamber with 210 mM KCl *cis* and 210 mM CaCl$_2$ *trans* as the source of permeant ions. (D) Current-voltage relationship with KCl and CaCl$_2$ as the source of permeant ions. Top, 210 mM CaCl$_2$ *cis* chamber, 210 mM KCl *trans* chamber; bottom, 210 mM KCl *cis* chamber, 210 mM CaCl$_2$ *trans* chamber. Data were displayed as mean ± SD from 3 or 4 independent experimental recordings. Source data are available online for this figure.

intoxication, sp-Ssp4 was expressed in *E. coli* and ROS levels quantified by OxyBURST Green staining. A significant increase in staining was recorded 2 and 3 h after induction of sp-Ssp4, indicating that the cells had entered a state of oxidative stress, but this was prevented by co-expression of Sip4 (Fig. 6A,B). In contrast, no specific increase in OxyBURST staining was observed when *E. coli* was intoxicated with sp-Ssp6 or the cytoplasmic-acting effector Ssp5 (Fig. 6C), suggesting that the generation of ROS is a specific consequence of Ssp4 intoxication.

The generation of ROS can be triggered via a number of different pathways, and several gene deletion mutants have been reported to show reduced susceptibility to antibiotics or toxins as a result of reduced ROS production (Cudic et al, 2017; Dong et al, 2015; Kohanski et al, 2008; Mahoney and Silhavy, 2013). As Ssp4 targets the membrane, we investigated the potential contribution of the CpxAR-dependent envelope stress response, which is triggered in response to membrane disruption and has been implicated in ROS-dependent toxicity (Cudic et al, 2017; Kohanski et al, 2008; Mahoney and Silhavy, 2013). The sp-Ssp4 protein was expressed in *E. coli* BW25113 Δ*cpxA* and Δ*cpxR* mutants, as well as in a Δ*lacA* control mutant derived from the same collection (Baba et al, 2006). The Δ*cpxA* mutant showed a modest decrease in OxyBURST signal intensity upon Ssp4 intoxication, indicating that CpxA activation may partly contribute to Ssp4-induced ROS generation (Fig. EV1A). However, this effect was not sufficient to provide the Δ*cpxA* mutant with measurable resistance against Ssp4 in the context of T6SS-mediated competition (Fig. EV1B), while the Δ*cpxR* mutant showed similar resistance towards Ssp4 as the Δ*lacA* control in both assays.

## Use of Tn-seq to identify mutants resistant to individual effectors reveals that MucA disruption protects *P. fluorescens* against T6SS attack

We next aimed to identify mutations conferring resistance to Ssp4 intoxication in an unbiased manner using transposon insertion site sequencing (Tn-seq) (Fig. 7A). A transposon insertion library of *P. fluorescens* was co-cultured with *S. marcescens* Db10 Δ9 or Δ9 + *ssp4*. The recovered *P. fluorescens* population was then subjected to Tn-seq in order to identify genes whose inactivation led to an over-representation of the corresponding mutants in the *P. fluorescens* population exposed to Ssp4 delivery compared with the population exposed to the Δ9 control. The Δ9 + *ssp6* strain was also included, and the equivalent comparison with Δ9 was performed, as an example of a pore-forming effector which does not detectably harm *P. fluorescens*. The Tn-seq experiment revealed one gene whose disruption resulted in increased relative survival under conditions of Ssp4 delivery, *moaC*, and two genes whose disruption resulted in increased relative survival under Ssp6 delivery, *xcpW* and *mucA*

(FDR <0.05, Figs. 7B and EV2A; Table 1; Dataset EV1). MoaC is required for molybdenum cofactor biosynthesis (Leimkuhler and Iobbi-Nivol, 2016), XcpW is a minor pseudopilin in the Type II secretion system (Zhang et al, 2018), and MucA is a negative regulator of the sigma factor AlgU, preventing activation of the AlgU regulon in response to cell envelope stress (Schofield et al, 2021). A limited number of genes whose disruption resulted in decreased relative survival under Ssp4 intoxication conditions were identified (Table 1); the two most strongly affected of these, *01041* and *00266*, encoding proteins of unknown function, were selected for further study, along with *moaC*, *xcpW* and *mucA*. Transposon insertions were observed across the full length of *moaC*, *xcpW*, *01041* and *00266* (Fig. EV2B). However, the first 200 bp of *mucA* was devoid of transposon insertions, suggesting that this region of the gene is required for bacterial viability, consistent with the observation that the N-terminal AlgU binding domain of MucA is essential for viability in *P. aeruginosa* (Schofield et al, 2021).

To validate the results of the Tn-seq experiment, mutants with in-frame deletions of *moaC*, *xcpW*, *01041*, *00266* and the region encoding the C-terminal 120 amino acids of MucA (*mucA*$_{\Delta76-195}$) were assessed for susceptibility to Ssp4 and Ssp6 in the standard co-culture assay. The *mucA*$_{\Delta76-195}$ mutant displayed a 2–3 log$_{10}$ increase in survival compared with wild type *P. fluorescens* against Δ9 + *ssp4* (Fig. 7C). This finding prompted us to re-examine the Tn-Seq data and, indeed, *mucA* was the third most increased gene in terms of number of insertions in Δ9 + *ssp4* vs Δ9 (2.7x, *p* < 0.001) but missed the FDR < 0.05 cut-off in this case (Dataset EV1; Fig. EV2B). Recovery of the Δ*moaC* mutant was significantly lower than the wild type under all conditions except when exposed to Δ9 + *ssp4*, indicating this mutation may confer a slight fitness advantage under conditions of Ssp4 intoxication (Fig. 7D). The other *P. fluorescens* mutants showed very small or no differences in resistance to Ssp4 and Ssp6 intoxication (Appendix Fig. S6).

To determine if the increased resistance of *mucA*$_{\Delta76-195}$ was specific to intoxication by Ssp4, this mutant was competed against wild-type *S. marcescens* Db10, with a full set of T6SS effectors, and strains delivering unrelated individual effectors, namely the peptidoglycan amidase Ssp2 (Δ9 + *ssp2*) and the DNase Rhs2 (Δ9 + *rhs2*). The *mucA*$_{\Delta76-195}$ mutant was considerably more resistant than wild-type *P. fluorescens* against all the effector-delivering strains, demonstrating that it is resistant to T6SS attack rather than specifically resistant to the action of one effector (Fig. 7E,F). When the Δ*moaC* mutant was co-cultured with wild type and T6SS-inactive Db10, the slight fitness disadvantage observed in the no-attacker control above was not replicated. However, recovery of the Δ*moaC* mutant was significantly higher than that of wild type *P. fluorescens* when competed with wild type but not T6SS-inactive Db10 (Fig. 7E). These data are consistent with a small but significant benefit of *moaC* deletion when exposed to T6SS attack.

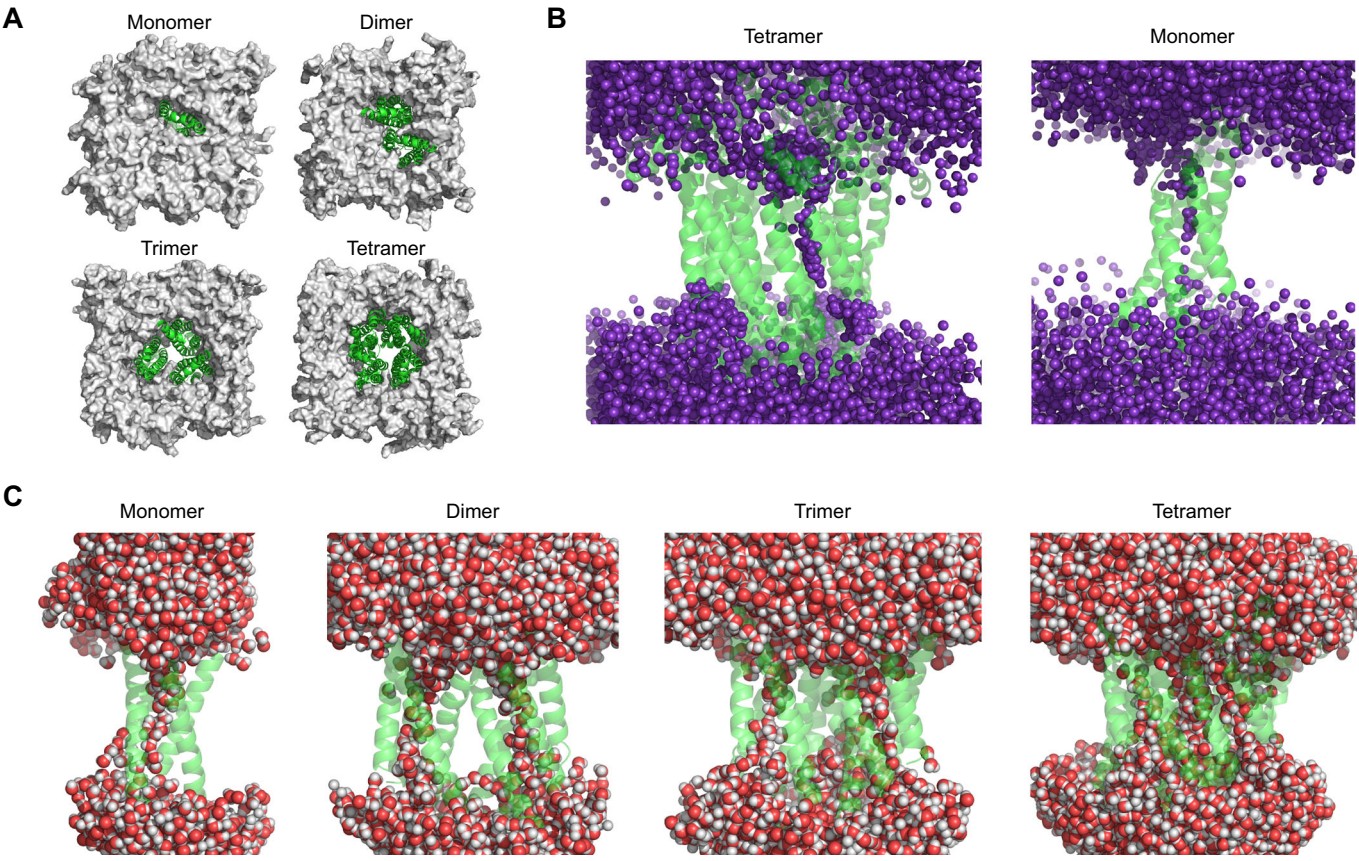

**Figure 5. Ssp4 stably inserts into a model membrane in various oligomeric states in which each subunit forms hydrated pores across the membrane.**

Molecular dynamics simulations were used to assess stability, hydration and ion permeation of oligomeric Ssp4 pores. (A) Model structures of the Ssp4$_{114-302}$ monomer, dimer, trimer and tetramer in a POPE lipid bilayer. The structures are viewed from the side on which the N- and C-termini of Ssp4 are located, predicted to be on the periplasmic side of the membrane in vivo. (B) Overlays of K$^+$ ion positions from a 1 μs simulation of the tetrameric assembly (left) or the monomer (right) show the pathway of K$^+$ conduction observed in Ssp4; only one subunit actively contributes to current at a time in the tetramer. K$^+$ ions are shown in purple. (C) Hydration of the stable monomer, dimer, trimer and tetramer Ssp4$_{114-302}$ assemblies in the membrane showing water molecules (red and grey) forming hydrated pores.

## Ssp4 is very common in *Serratia* and represents a new family of widely occurring effectors

We previously reported that homologues of Ssp6 can be found across the *Enterobacterales* but not outside this order (Mariano et al, 2019). Given the differences between Ssp4 and Ssp6, including target species specificity, we investigated the distribution of Ssp4-like proteins. Interrogation of the previously-reported pan-genome of the genus *Serratia* (Williams et al, 2022) revealed that Ssp4 is more widely distributed across the genus and occurs much more frequently than Ssp6 (Fig. 8A), which may be a consequence of its greater efficacy (Fig. 2). Given this level of conservation in *Serratia*, and the polyphyletic pattern of occurrence across the genus, we were interested to see how widely Ssp4 is found in other bacterial species. Using HMMER homology searching, we identified Ssp4 homologues across the *Gammaproteobacteria* (Fig. 8B; Dataset EV2; Appendix Fig. S7). Ssp4 homologues in a number of genera within the order *Enterobacterales* formed a group closely related to Ssp4 in *S. marcescens*, whilst two other groups of more-distantly related Ssp4-like proteins were observed, one containing a large number of Ssp4 homologues across the genus *Pseudomonas*, and

the other containing Ssp4 homologues in several orders of marine bacteria (Fig. 8C). Examining the genomic context of these Ssp4 homologues showed that Ssp4 genes are encoded next to a small downstream gene that is likely to encode a cognate immunity protein (Fig. 8D). Additionally, for all three groups of Ssp4-like proteins, whilst the corresponding genes are often found in loci distant from any T6SS genes, in some cases they are found next to genes encoding T6SS components which recruit effectors (Hcp and VgrG) or within large T6SS gene clusters (Fig. 8D), strongly indicating that these groups of Ssp4-like proteins are also T6SS-dependent effectors. Overall, this analysis indicates that Ssp4 is the founding member of a new family of ion-selective pore-forming effectors, which contains at least three groups spanning a number of bacterial orders.

## Discussion

In this study, we have shown that Ssp4, an effector delivered by the T6SS of *S. marcescens* Db10, is a potent anti-bacterial toxin that forms ion-selective pores in the inner membrane of intoxicated

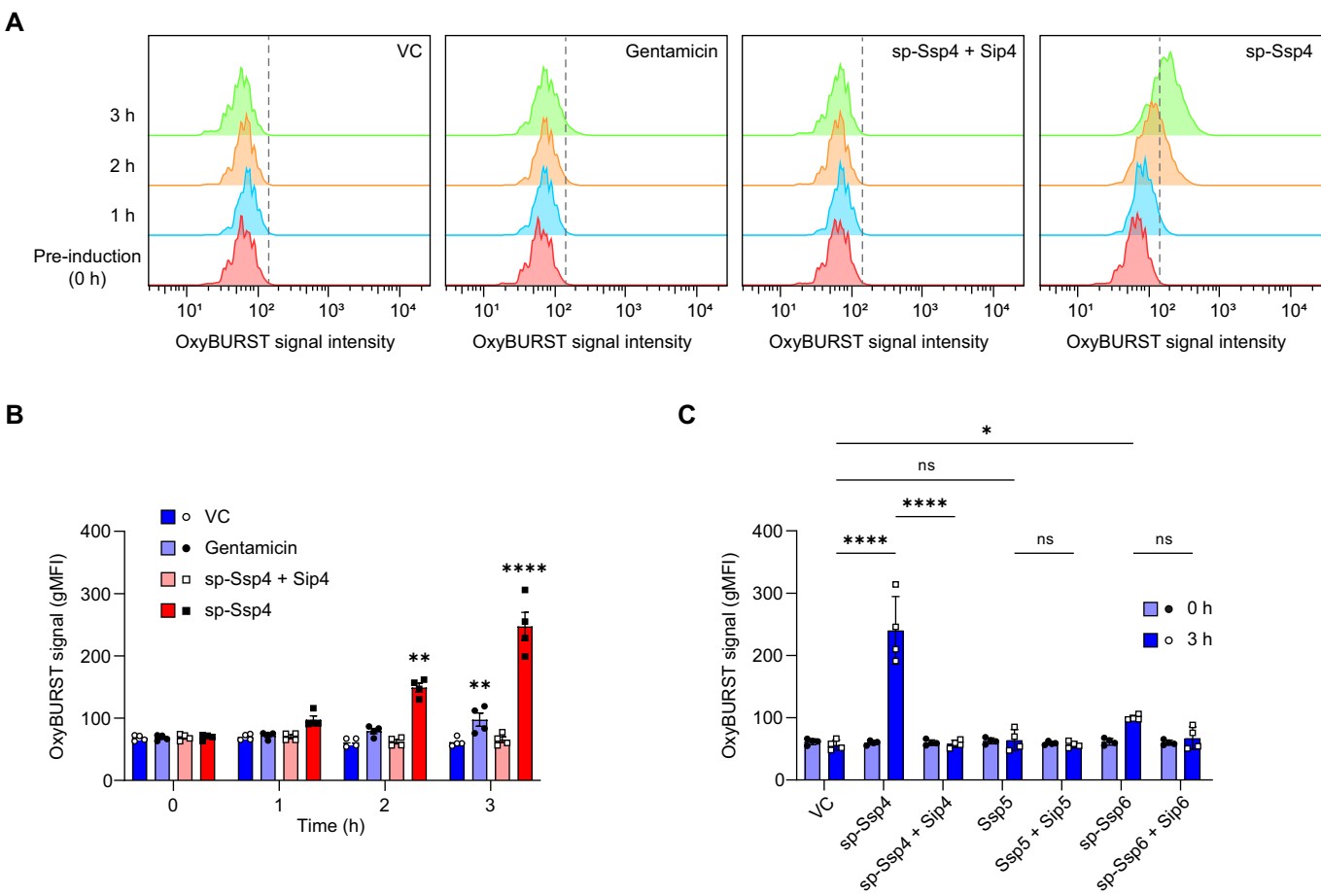

**Figure 6. Ssp4 intoxication is accompanied by an increase in the level of intracellular reactive oxygen species (ROS).**

(**A**) Representative flow cytometry histograms showing the OxyBURST Green signal intensity over time from cells of *E. coli* MG1655 carrying the vector control (VC, pBAD18-Kn) or plasmids directing the expression of Ssp4 fused with an N-terminal signal peptide (sp-Ssp4), either alone or with Sip4, following induction with 0.2% L-arabinose. Cells were separately exposed to 5 μg/ml gentamicin for comparison. (**B**) Quantification of OxyBURST Green signal from the experiment in panel a, with data presented as mean ± SEM, with individual data points overlaid (*n* = 4 biological replicates); ****P < 0.0001, **P < 0.01, compared with pre-induction levels; repeated measures ANOVA with Dunnett test. gMFI, geometric mean fluorescence intensity. (**C**) Quantification of OxyBURST Green intensity from *E. coli* MG1655 carrying plasmids directing expression of sp-Ssp4, Ssp5 and sp-Ssp6, with or without their cognate immunity proteins, pre-induction and following 3 h induction. Data were presented as mean ± SEM with individual data points overlaid (*n* = 4 biological replicates); ****P < 0.0001, *P < 0.05, ns not significant, one-way ANOVA with Tukey's test; for clarity, only selected comparisons are displayed. *P* values from left to right (**B**) *P* = 0.0025, *P* = 0.002, *P* < 0.0001; (**C**) *P* < 0.0001, *P* < 0.0001, *P* > 0.9999, *P* = 0.0308, *P* > 0.9999, *P* = 0.2662. Source data are available online for this figure.

cells, leading to a catastrophic disruption of their membrane potential. Homologues of Ssp4 are found widely in Gram-negative bacteria and represent a new family of T6SS effector proteins with at least three sub-groups.

The discovery that Ssp4 is a pore-forming effector that displays a preference for cations was somewhat unexpected, given that Db10 possesses another T6SS effector, Ssp6, which, although unrelated to Ssp4, is also a cation-selective pore-forming toxin and causes similar membrane depolarisation and cessation of growth in intoxicated cells. However, the two effectors are not redundant, and differ in their potency, ion selectivity, target species specificity, and the distribution of their homologues across bacterial genera. T6SS effectors are often considered to be 'broad spectrum' based on the fact that they target conserved, essential cellular structures and molecules in bacterial cells. However, here we observed that whilst Ssp4 was able to effectively intoxicate all species tested, Ssp6 could intoxicate *S. marcescens*, *E. coli*,

and *Ent. cloacae*, but was not able to intoxicate *P. fluorescens* or *B. thailandensis*, implying it may only work against species relatively closely related to *Serratia*. Several other studies have also reported examples of other T6SS effectors having species-specific activity (Hersch et al, 2020b; Kamal et al, 2020; Le et al, 2020; Santos et al, 2024). For example, *E. coli* is protected against the *V. cholerae* peptidoglycan hydrolase effector TseH by the induction of envelope stress responses, whereas *Aeromonas dhakensis* and *Edwardsiella tarda* are susceptible to T6SS delivery of this effector (Hersch et al, 2020b), whilst delivery of the DNA deaminase effector DddA leads to inhibition of growth in some species but only accumulation of C → T mutations in others (de Moraes et al, 2021). Interestingly, whilst homologues of Ssp6 are restricted to the Enterobacterales, Ssp4-like proteins are found much more widely, including in *Pseudomonas* and *Vibrio* species. Ssp4 is also much more commonly found across *Serratia* than Ssp6. Taken together, we suggest that Ssp4-type effectors

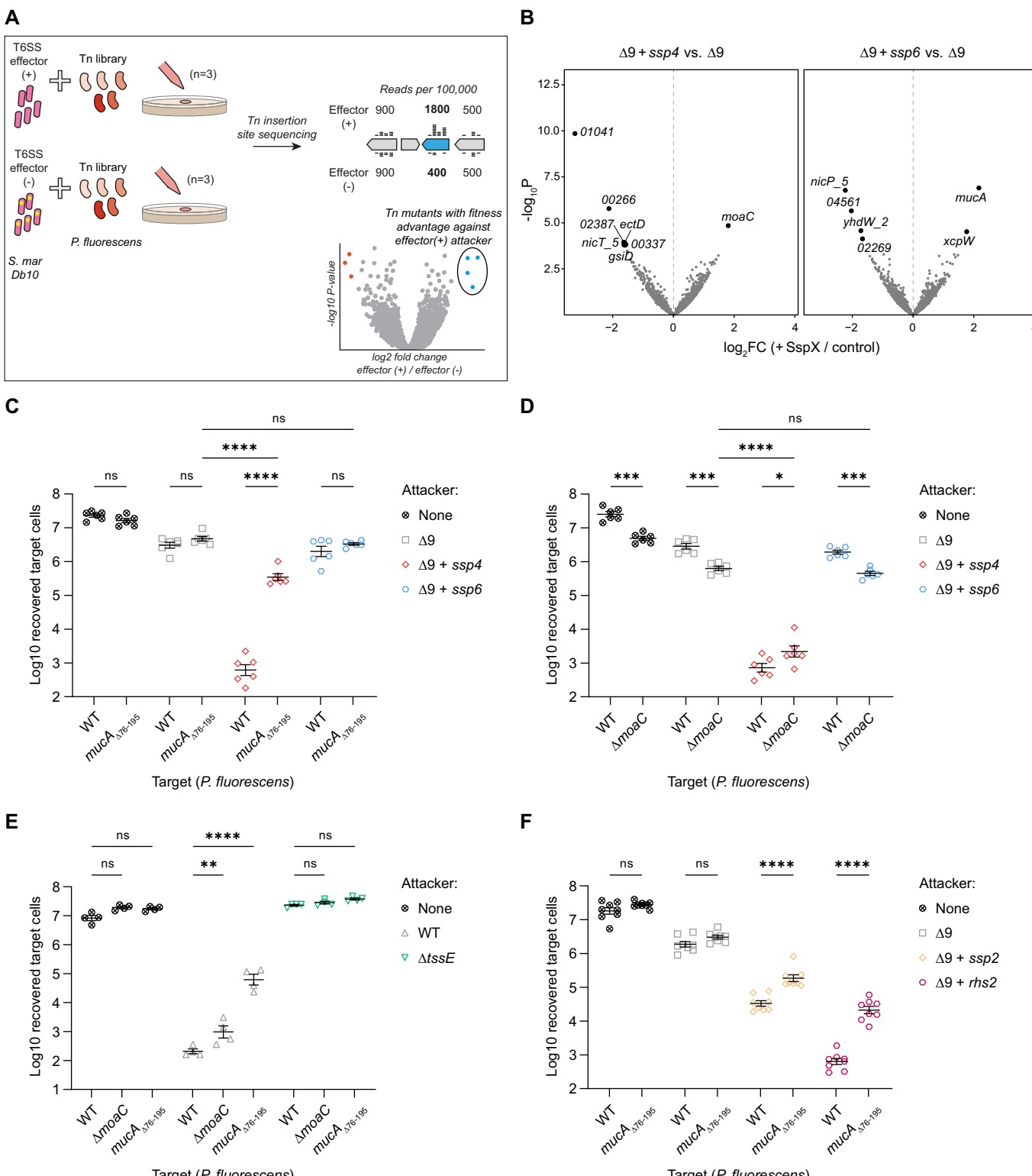

are commonly used for competition between more-distantly related species, whilst Ssp6 is used only for intra-order competition within the Enterobacterales. This broader specificity, coupled with the higher potency of Ssp4, may explain why Ssp4 is found so frequently in *S. marcescens*. However, this frequency also means it is less useful for

competition within species, hence certain isolates gain a competitive intra-species advantage by horizontal acquisition of the rarer Ssp6.

Structural prediction reveals that Ssp4 is an α-pore-forming toxin (α-PFT), a class which includes several colicins (Ulhuq and Mariano, 2022) and likely other T6SS pore-forming effectors,

**Figure 7.　Use of Tn-seq to identify mutants resistant to Ssp4 or Ssp6 reveals that MucA disruption protects *P. fluorescens* against the T6SS.**

(A) Schematic illustration of the Tn-seq experiment. *S. marcescens* Db10 lacking known anti-bacterial effectors (effector (−), strain Δ9) or delivering only one effector (effector (+), strains Δ9 + *ssp4* or Δ9 + *ssp6*) was co-cultured with a saturated transposon insertion library of *P. fluorescens* 55 for 4 h. Genomic DNA from the recovered total population was subjected to Illumina sequencing using transposon specific primers and the number of reads mapping to each *P. fluorescens* gene determined, indicating the relative abundance of *P. fluorescens* mutants with insertions in each gene within the final population. Comparing the relative abundance of mutants with Tn insertions in particular genes between co-culture with effector (+) vs. effector (−) attackers, by determining the fold change in normalised Tn insertion frequency between effector (+) and (−) for each gene, those genes whose disruption results in a fitness advantage against T6SS-mediated intoxication by the effector can be identified (blue points). (B) Volcano plots summarising the change in recovery of transposon insertion mutants between control (Δ9) and Ssp4-delivering (left) or Ssp6-delivering (right) attackers, on a per gene basis. Log2 fold change in normalised read count is plotted against −log10 $p$ value, and genes significantly altered between conditions are highlighted (FDR <0.05, $n = 3$ biological replicates). (C–F) Recovery of wild type (WT) or defined mutants (*mucA*$_{\Delta 76-195}$ or Δ*moaC*) of *P. fluorescens* following co-culture with attacking strains of *S. marcescens* Db10 as indicated (where Δ9 + *ssp2* and Δ9 + *rhs2* intoxicate using only Ssp2 or Rhs2, respectively). None, no-attacker. Data were presented as mean ± SEM with individual data points overlaid ($n = 6$, 4 or 8 biological replicates for panels **C–F**, respectively); ****$P < 0.0001$, ***$P < 0.001$, *$P < 0.05$, ns not significant; one-way ANOVA with Tukey's test; for clarity, only selected comparisons are displayed. $P$ values from left to right (C) $P = 0.9648$, $P = 0.8699$, $P < 0.0001$, $P < 0.0001$, $P = 0.9516$, $P = 0.7689$; (D) $P = 0.0001$, $P = 0.0003$, $P < 0.0001$, $P = 0.0182$, $P = 0.9539$, $P = 0.0007$; (E) $P = 0.3004$, $P = 0.4527$, $P = 0.0034$, $P < 0.0001$, $P = 0.9994$, $P = 0.8675$; (F) $P = 0.7814$, $P = 0.6510$, $P < 0.0001$, $P < 0.0001$. Panels (C, D) form part of the larger experiment depicted in Appendix Fig. S6. Source data are available online for this figure.

including Ssp6, Tse5 and Tse4 from *P. aeruginosa*, and VasX from *V. cholerae* (Gonzalez-Magana et al, 2022; LaCourse et al, 2018; Mariano et al, 2019; Miyata et al, 2013). Upon delivery to their target cellular compartment, α-PFT monomers typically undergo a conformational change resulting in exposure of hydrophobic or amphipathic helices, followed by oligomerisation and concomitant membrane insertion to form the mature pore (Ulhuq & Mariano, 2022). Such toxins are thus well suited to delivery by the T6SS, where two forms of the protein are likely to exist: a pre-secretion form, which is soluble and compact to allow loading into the Hcp tube, and an active form with exposed hydrophobic helices primed for membrane insertion. We believe that our AlphaFold-generated structures represent the active/membrane-inserted conformation of Ssp4, given their exposed transmembrane helices and predicted abilities to insert in a membrane and conduct ions. We have previously observed stabilisation of Ssp4 by Hcp, suggesting that an alternative, soluble pre-secretion form of Ssp4 is stabilised by a chaperone-like interaction with Hcp (Cianfanelli et al, 2016). On the other hand, the basis for the resistance to Ssp6 intoxication observed in *P. fluorescens* remains to be elucidated. Differences in lipid composition or membrane structure could mean that membrane insertion and oligomerisation of Ssp6 cannot proceed properly to form a mature pore.

To date, no structures have been reported for membrane pores formed by T6SS-delivered effectors. Here, we present the first structural prediction for such a pore, supported by molecular dynamics simulations and comparison with experimental electrophysiology data. Our modelling suggests that individual membrane-inserted Ssp4 monomers contain a water-filled, ion-conducting channel and that Ssp4 may be able to adopt several oligomeric states (monomer, dimer, trimer and tetramer) with the ability to reside stably in a membrane and conduct ions. The mechanism of ion conductance is not dependent on oligomeric state, since the ions do not flow through a central channel formed by subunit oligomerisation. Interestingly, even in multimeric assemblies, it appears that only one subunit of Ssp4 actively conducts ions at a time. Our data do not allow us to predict in which of these oligomeric states Ssp4 exists in vivo. Indeed, several states may co-exist, or the preferred state may vary with membrane or environmental conditions. Alternatively, the simplest possibility, that Ssp4 can act as a monomer, may present the most efficient mechanism of toxicity. Nevertheless, our basic model of

conductance through the monomeric unit of Ssp4 is supported by calculated and experimental conductance values falling within a similar range. We note that the simulated values were generated using a lower ion concentration, 150 mM KCl, than the experimental values (510 and 210 mM KCl), which could lead to lower conductance. However, the very similar experimental values obtained under symmetric and asymmetric conditions (Fig. 4B) imply that we are close to the maximal single-channel conductance at 210 mM and conductance at 150 mM is unlikely to be greatly different. The formation of a multimeric pore would require multiple Ssp4 proteins to be delivered into a recipient cell, most likely simultaneously. Ssp4 is an Hcp-dependent cargo effector (Cianfanelli et al, 2016), meaning that it interacts with one or two hexameric rings of Hcp inside the T6SS-delivered puncturing structure. Given that the Hcp tube may comprise over 200 rings of Hcp (Wang et al, 2017), it is likely that many individual Ssp4 proteins are delivered per firing event, allowing several multimeric pores to form in the recipient cell, even from just one shot. In contrast, if Ssp4 were delivered by association with VgrG or PAAR, several shots might be required, illustrating the potential importance of delivery mode to effector function.

We have demonstrated experimentally that, in common with Ssp6, Tse5, and most likely Tse4 (Gonzalez-Magana et al, 2022; LaCourse et al, 2018; Mariano et al, 2019), Ssp4 forms ion-conducting pores in target cell membranes. This leads to unregulated movement of ions across the inner membrane and disrupted membrane potential, in turn interfering with the proton motive force, ATP synthesis, membrane transport and a variety of other cellular processes (Benarroch and Asally, 2020). These effectors cause membrane depolarisation without affecting the overall integrity of the inner membrane. In contrast, several T6SS effectors cause permeabilisation and loss of membrane integrity, suggesting they may form larger, non-specific pores, namely VasX (Miyata et al, 2013) and Tme1/2 from *V. parahaemolyticus* (Fridman et al, 2020). Whilst Ssp4 does not cause lysis or loss of membrane integrity, it can cause terminal inhibition of growth and loss of viability, as shown by the number of surviving viable target cells being much lower than the initial inoculum in co-culture assays (Fig. 2A,B).

Of the pore-forming T6SS effectors described to date, ion selectivity has only been determined in vitro for Ssp4, Ssp6 and Tse5, whilst cell-based assays suggested that Tse4 forms pores

**Table 1.** Genes of *P. fluorescens* 55 in which mutants carrying transposon insertions were significantly enriched or depleted in the presence of *S. marcescens* delivering Ssp4 or Ssp6.

| Identifier | Gene name | Product | Comparison | Log2FC | -Log10 P value | FDR |
|---|---|---|---|---|---|---|
| 33931E_Pfluorescens55_01260 | *moaC* | Cyclic pyranopterin monophosphate synthase | Δ9 + *ssp4* vs. Δ9 | 1.80 | 4.85 | 1.15E-02 |
| 33931E_Pfluorescens55_00337 | *00337* | Hypothetical protein | | −1.56 | 3.82 | 4.97E-02 |
| 33931E_Pfluorescens55_03917 | *ectD* | Ectoine dioxygenase | | −1.60 | 3.92 | 4.97E-02 |
| 33931E_Pfluorescens55_02472 | *gsiD* | Glutathione transport system permease protein | | −1.61 | 3.79 | 4.97E-02 |
| 33931E_Pfluorescens55_03709 | *nicT5* | Putative metabolite transport protein NicT | | −1.63 | 3.82 | 4.97E-02 |
| 33931E_Pfluorescens55_02387 | *02387* | Hypothetical protein | | −1.63 | 3.95 | 4.97E-02 |
| 33931E_Pfluorescens55_00266 | *00266* | Hypothetical protein | | −2.12 | 5.77 | 2.04E-03 |
| 33931E_Pfluorescens55_01041 | *01041* | Hypothetical protein | | −3.24 | 9.85 | 3.41E-07 |
| 33931E_Pfluorescens55_01704 | *mucA* | Sigma factor AlgU negative regulatory protein | Δ9 + *ssp6* vs. Δ9 | 2.17 | 6.90 | 2.07E-04 |
| 33931E_Pfluorescens55_02495 | *xcpW* | Type II secretion system protein J | | 1.77 | 4.52 | 1.47E-02 |
| 33931E_Pfluorescens55_02269 | *02269* | Hypothetical protein | | −1.67 | 4.13 | 2.98E-02 |
| 33931E_Pfluorescens55_04780 | *yhdW2* | Putative amino acid ABC transporter-binding protein | | −1.72 | 4.58 | 1.47E-02 |
| 33931E_Pfluorescens55_04561 | *04561* | Hypothetical protein | | −2.03 | 5.65 | 1.81E-03 |
| 33931E_Pfluorescens55_03230 | *nicP5* | Porin-like protein NicP | | −2.23 | 6.77 | 2.07E-04 |

Genes are included if the number of sequencing reads derived from transposon insertions within the gene was significantly altered between the attacking strains compared, based on an FDR value <0.05. The table summarises data from three biologically independent co-cultures between a *P. fluorescens* 55 transposon insertion library and *S. marcescens* strains Δ9 (lacking known anti-bacterial effectors), Δ9 + *ssp4* (delivering Ssp4) or Δ9 + *ssp6* (delivering Ssp6).
FC fold change, FDR false discovery rate.

specific for monovalent cations (Gonzalez-Magana et al, 2022; LaCourse et al, 2018; Mariano et al, 2019). It is interesting to note that all these effectors form pores with a preference for cations, although Ssp6 appears to be exclusive for cations, whilst Ssp4, like Tse5, is also able to transport anions. The ability of Ssp4 to conduct anions as well as cations may contribute to its greater toxicity in vivo, although the ionic gradient(s) dissipated by all these toxins will vary with the extracellular environment of the intoxicated cell. Intriguingly, our data also revealed that the activity of the Ssp4 pore may be affected by the presence of $Ca^{2+}$ at the cytoplasmic face of the inner membrane, since the properties of the pore changed when $Ca^{2+}$ ions were present in the *trans* chamber. Whilst it is widely reported that $Ca^{2+}$ does not interact strongly with neutral phosphatidylethanolamine (PE) lipids directly, it has been suggested that $Ca^{2+}$ binds to phosphate groups of all phospholipids, independent of their charge (Huster et al, 2000). It is possible that this could have local effects at the membrane level, resulting in altered gating or conductance properties of Ssp4. We predict that the *trans* chamber is equivalent to the cytoplasm in our system, since Ssp4 acts, and therefore must enter the inner membrane, from the periplasm in vivo, which corresponds to its addition to the lipid bilayer on the *cis* side in vitro. $Ca^{2+}$ has been shown to modulate the transport properties of Tse5 and the SARS-CoV-2 E protein, suggesting that regulation by $Ca^{2+}$ may be a common property of microbial pore-forming toxins (Antonides et al, 2022; Rojas-Palomino et al, 2024).

Another interesting difference between the two pore-forming effectors deployed by *S. marcescens* Db10 is that intoxication by Ssp4 but not Ssp6 leads to the accumulation of intracellular ROS in *E. coli*. Accumulation of ROS causes damage to DNA, protein and lipids and can contribute to the lethality of antibiotics under certain conditions (Dwyer et al, 2014; Hong et al, 2019). It is possible that ROS production also contributes to the higher level of toxicity

observed for Ssp4 compared with Ssp6. The reason for the difference between the effectors is not clear, but might perhaps reflect the ability of Ssp4 to interfere with a greater number of ion gradients. Bacteria possess multiple protective pathways that counter ROS damage. Disruption of one of these, SoxRS, resulted in increased susceptibility of *E. coli* to T6SS attacks, suggesting that ROS generation can contribute to T6SS-mediated anti-bacterial activity (Dong et al, 2015). Our observation that intoxication with Ssp4, but not Ssp5 or Ssp6, causes increased intracellular ROS suggests that ROS production may be a specific consequence of certain effectors rather than a general response to the stress induced by T6SS attack. Activation of CpxA in response to cell envelope stress has been implicated in aminoglycoside-induced ROS production, and its loss can increase resistance towards such compounds (Kohanski et al, 2008). Deletion of *cpxA* caused only a modest reduction in ROS accumulation following expression of sp-Ssp,4 suggesting that ROS production in response to Ssp4 is largely independent of the Cpx pathway. Loss of CpxA had no impact on *E. coli* susceptibility to Ssp4 in co-culture, indicating that the Cpx stress response pathway does not significantly potentiate or protect against T6SS-delivered Ssp4, consistent with observations for TseH of *V. cholerae* (Hersch et al, 2020b).

The strong Ssp4-dependent anti-bacterial activity observed in co-culture experiments provided an opportunity to investigate whether an unbiased approach, Tn-seq, could identify mutants resistant to a pore-forming effector. One gene whose inactivation caused a modest level of resistance to Ssp4, *moaC*, was identified, although there is a trade-off with the fact that loss of MoaC, a protein required for molybdenum cofactor (Moco) biosynthesis, has a deleterious impact on overall fitness. It is currently unclear why interruption of Moco synthesis or loss of the ability to assemble Moco-containing enzymes, which are redox-active enzymes involved in electron-transfer reactions during nitrogen,

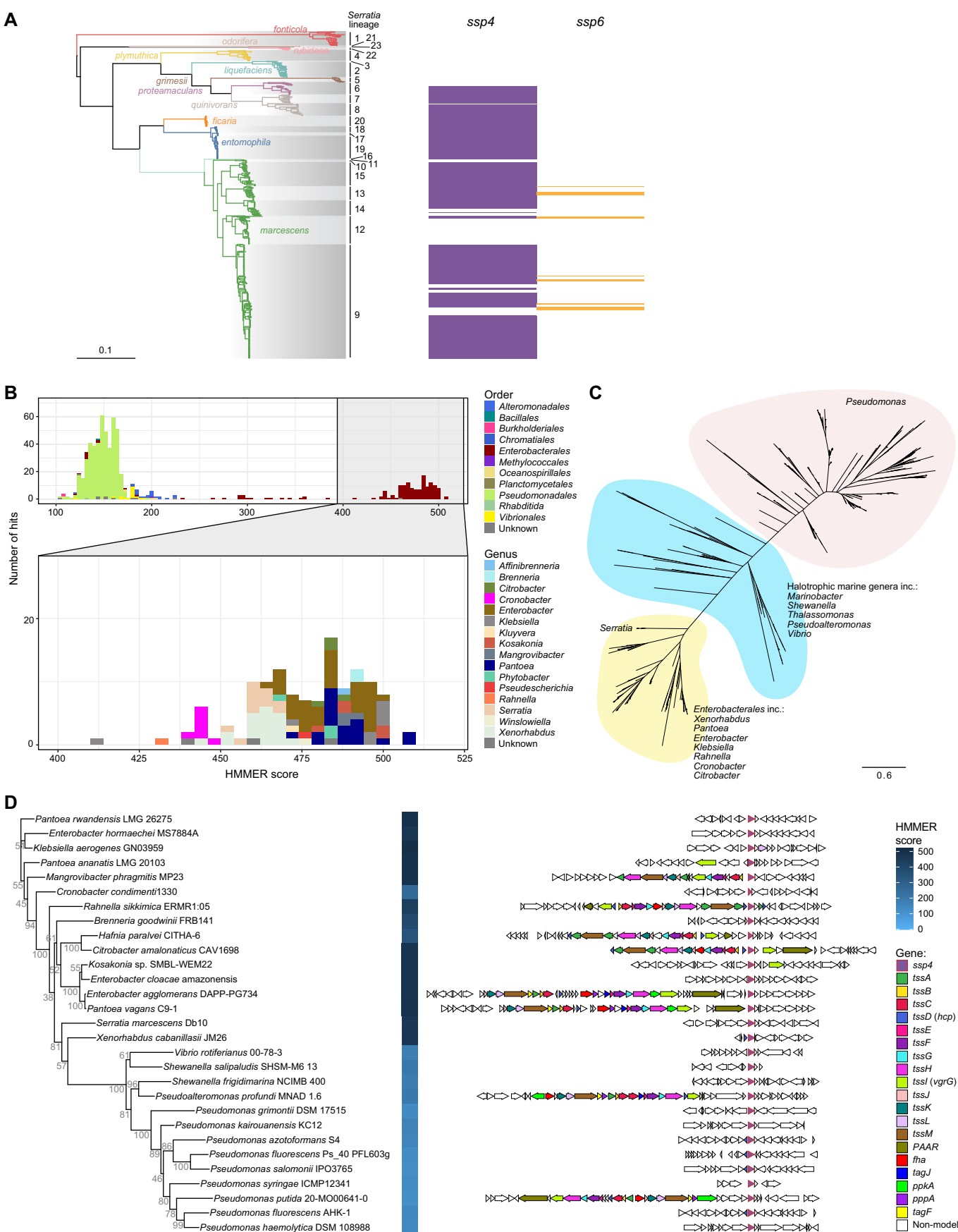

**Figure 8.  Ssp4-like proteins occur very frequently in *Serratia* and widely across other genera.**

(A) Presence/absence of the *ssp4* gene (purple) and the *ssp6* gene (orange) shown alongside a maximum-likelihood phylogenetic tree of 664 *Serratia* genomes determined previously from a core gene alignment (Williams et al, 2022). (B) Homologues of Ssp4 identified using HMMER homology searching, coloured by Order of the encoding bacteria (top panel) or by Genus (bottom/expanded panel). (C) Unrooted maximum-likelihood phylogeny of all identified Ssp4 homologues showing the three groups formed by Ssp4-like proteins in *Pseudomonas* (pink), marine genera (cyan), and Enterobacterales (yellow). (D) Maximum-likelihood tree of selected Ssp4 protein sequences from (C), with hmmsearch score (left) and the genetic context of the corresponding encoding gene (right). Bootstrap values are indicated on the tree and the scale indicates number of substitutions per site. Conserved T6SS genes and *ssp4* are coloured as per the key, with other genes white.

carbon, and sulphur metabolism (Leimkuhler and Iobbi-Nivol, 2016), would provide an advantage under conditions of Ssp4 intoxication. One hypothesis is that eliminating the activity of these enzymes in the context of elevated oxidative stress could be beneficial. MoaC acts before the step in Moco synthesis requiring the activity of two proteins outside the Moco-specific pathway, IscS and TusA. Hence, loss of competition from MoaC could free these enzymes to aid in repairing oxidative damage to Fe-S clusters in other proteins. However, any advantage of *moaC* inactivation will only be seen in conditions when Moco-dependent electron-transfer reactions are not essential, such as during growth on LB.

More strikingly, the Tn-seq approach revealed that loss of the C-terminal domain of the anti-sigma factor MucA provides *P. fluorescens* with substantially increased protection against overall T6SS attack. Given that Ssp6 intoxication does not cause a detectable phenotype in *P. fluorescens*, the selection of *mucA* mutants in the Tn-seq comparison of Δ9 + *ssp6* vs Δ9 was unexpected. It is possible that delivery of Ssp6 into *P. fluorescens* leads to rare or immature pores, conferring a small fitness defect which generates a selection pressure for resistance to T6SS attack, which is detectable by this assay. Alternatively, reintroduction of Ssp6 into the Δ9 mutant might increase T6SS firing rate via a checkpoint for effector loading (Liang et al, 2019), leading to increased T6SS damage to the target cells, either from penetration or from the delivery of an as-yet-unidentified effector. In both cases, a small fitness advantage caused by truncation of MucA in the presence of Δ9 + *ssp6* may only be apparent when the *mucA* mutant is directly competing against other genotypes of *P. fluorescens*, as is the case during the Tn-seq selection.

MucA has been extensively characterised in *P. aeruginosa* due to its role in regulating production of the extracellular polysaccharide (EPS) alginate and the fact that mutations causing C-terminal truncations of MucA lead to a mucoid (alginate overproducing) phenotype in isolates from cystic fibrosis patients (Boucher et al, 1997; Schofield et al, 2021). Production of alginate in *P. fluorescens* is also controlled by MucA, and truncations of MucA lead to EPS production, in concert with other changes in gene expression consistent with its role as an anti-sigma factor (Borgos et al, 2013). Previous studies have shown that production or overproduction of EPS, either as a membrane-bound capsule or free EPS, by *V. cholerae*, *E. coli* and *Klebsiella* can provide protection against T6SS attack (Flaugnatti et al, 2021; Granato et al, 2023; Hersch et al, 2020b; Toska et al, 2018). This protection operates by providing a physical barrier and, in the case of free EPS, by facilitating spatial segregation of producers and bystanders away from T6SS-wielding attacker cells (Granato et al, 2023). Thus, we initially believed that the increased survival of the *P. fluorescens mucA*$_{\Delta76-195}$ mutant was due to increased alginate production, providing a physical barrier between attacker and target cells and promoting the

formation of *P. fluorescens* microcolonies, which spatially segregate the cells within them from *S. marcescens* attackers. However, deleting the alginate biosynthetic gene *algD* in the *mucA*$_{\Delta76-195}$ mutant, to prevent alginate production, did not reverse this protection (Fig. EV3). This implies that the MucA mutant is protected from T6SS attacks by a distinct resistance mechanism, not via physical protection provided by EPS overproduction. The nature of this mechanism remains to be elucidated but it may be linked with alginate-independent transcriptional and metabolic changes observed in a MucA mutant of *P. fluorescens*, including reorganisation of carbon utilisation and energy generation, reduced respiration, and upregulation of ribosomal protein expression (Borgos et al, 2013; Lien et al, 2015), and is likely also distinct from resistance resulting from stress responses to T6SS-inflicted damage (Hersch et al, 2020a).

In conclusion, Ssp4 represents the founding member of a widely distributed family of T6SS-dependent effectors which form ion-selective transmembrane pores. Ssp4 is one of two unrelated pore-forming effectors delivered by the T6SS of *S. marcescens* Db10, which differ in their ion selectivity, target specificity and impact on intoxicated cells. Moreover, these two effectors are likely to be used differently, one for 'intra-order' competition within closely related species, the other primarily between more-distantly related competitors. Co-occurrence of two such pore-forming effectors may be common (e.g. Tse5 and Tse4 in *P. aeruginosa*), and these effectors likely play an important synergistic role in overall T6SS-mediated anti-bacterial activity, as loss of membrane potential will impair the ability of intoxicated cells to repair damage or reverse nucleotide depletion caused by other types of effectors. Conversely, diverse mechanisms can provide targeted cells with some level of protection against T6SS attacks, including physiological changes induced upon truncation of MucA in *P. fluorescens*. Overall, this study provides further evidence for the diversity of effectors and complexity of competitive inter-bacterial interactions mediated by the T6SS.

## Methods

**Reagents and tools table**

| Reagent/resource | Reference or source | Identifier or catalogue number |
|---|---|---|
| **Experimental models** | | |
| Bacterial strains | This study | Appendix Table S2 |
| *Pseudomonas fluorescens* 55 transposon insertion library | This study | |

| Reagent/resource | Reference or source | Identifier or catalogue number |
|---|---|---|
| **Recombinant DNA** | | |
| Plasmids | This study | Appendix Table S2 |
| **Antibodies** | | |
| HRP-conjugated goat anti-mouse IgG | Bio-Rad | 170-6516 |
| Mouse anti-FLAG antibody | Sigma-Aldrich | F3165 |
| **Oligonucleotides and other sequence-based reagents** | | |
| Oligonucleotide primers | This study | Appendix Table S3 |
| **Chemicals, enzymes and other reagents** | | |
| Tryptone | Thermo Fisher Scientific | LP0042 |
| Yeast extract | Thermo Fisher Scientific | LP0021 |
| Select agar | Thermo Fisher Scientific | 30391023 |
| Ampicillin | Formedium | AMP25 |
| Kanamycin | Formedium | KAN0025 |
| Gentamicin | Invitrogen | 15750037 |
| Tetracycline | Sigma-Aldrich | T7660 |
| Chloramphenicol | Sigma-Aldrich | C0378 |
| Streptomycin | Scientific Laboratory Supplies | S6501 |
| L-rhamnose | Sigma-Aldrich | 83650 |
| L-arabinose | Thermo Fisher Scientific | A11921-30 |
| Isopropyl β-D-1-thiogalactopyranoside (IPTG) | Formedium | IPTG005 |
| Tris (hydroxymethyl) aminomethane | Formedium | TRIS01 |
| Glycine | Merck | G7126 |
| Sodium dodecyl (lauryl) sulfate (SDS) | Formedium | SDS0500 |
| Imidazole | Merck | 1047161000 |
| HEPES (N-(2-Hydroxyethyl) piperazine-N'-(2-ethanesulfonic acid)) | Formedium | HEPES10 |
| Ethylenediaminetetraacetic acid (EDTA) | Sigma-Aldrich | E6758 |
| Glycerol | VWR | 24388320 |
| β-mercaptoethanol | Sigma-Aldrich | M6250 |
| Dithiothreitol (DTT) | Formedium | DTT010 |
| Tris(2-carboxyethyl) phosphine hydrochloride (TCEP) | Apollo Scientific | BIT0122 |
| Bromophenol Blue | Sigma-Aldrich | 114391 |
| Instant Blue stain | Expedeon | ISB1L |
| Marvel Milk powder | Unico | XSP |
| Tween-20 | Sigma-Aldrich | P9416 |
| Millipore Immobilon™ Western Chemiluminescent HRP Substrate | Fisher Scientific | 11546345 |

| Reagent/resource | Reference or source | Identifier or catalogue number |
|---|---|---|
| Ni Sepharose | Cytiva | 17531806 |
| mPEG-MAL (methoxypolyethylene glycol maleimide) | Sigma-Aldrich | 63187 |
| Lysozyme | Sigma-Aldrich | L6876 |
| Benzonase Nuclease (Millipore) | Sigma-Aldrich | 70746-3 |
| cOmplete™, EDTA-free Protease Inhibitor Cocktail | Roche | 11873580001 |
| Invitrogen™ UltraPure™ Agarose | Thermo Fisher Scientific | 16500500 |
| Invitrogen™ DiBAC$_4$(3) (Bis-(1,3-Dibutylbarbituric Acid) Trimethine Oxonol) | Thermo Fisher Scientific | B438 |
| Propidium iodide (PI) | Thermo Fisher Scientific | P3566 |
| Melittin | Sigma-Aldrich | M2272 |
| Invitrogen™ OxyBURST™ Green H2DCFDA-SE | Thermo Fisher Scientific | D2935 |
| Formaldehyde 16% | Thermo Fisher Scientific | 28906 |
| Glutathione Sepharose 4B resin | Cytiva | 17075601 |
| PreScission Protease (Cytiva) | Sigma-Aldrich | GE27-0843-01 |
| Glutathione | Sigma-Aldrich | G4251 |
| Bovine phosphatidylethanolamine lipids (Avanti Research) | Merck | 840025 P |
| Phosphate Buffered Saline (PBS) | Fisher | 12559069 |
| Terminal Deoxynucleotidyl Transferase (TdT) | New England Biolabs (NEB) | M0315 |
| dCTP / ddCTP mix | New England Biolabs (NEB) | N0446S |
| Q5 Hot Start High-Fidelity DNA Polymerase | New England Biolabs (NEB) | M0493 |
| AMPure XP beads | Beckman Coulter | A63881 |
| NEBNext Multiplex Oligos for Illumina | New England Biolabs (NEB) | E7500 |
| **Software** | | |
| OMERO | http://openmicroscopy.org | |
| Tn-Seq Pre-Processor (TPP) tool in TRANSIT | DeJesus et al, 2015 | |
| featureCounts | Liao et al, 2014 https://subread.sourceforge.net/featureCounts.html | |
| MEMSAT-SVM (PSIPred) | http://bioinf.cs.ucl.ac.uk/psipred/ | |
| AlphaFold2 (Google Colab) | Jumper et al, 2021 https://colab.research.google.com | |
| AlphaFold2 (HPC, Dundee) | Jumper et al, 2021 University of Dundee HPC cluster | |
| PyMOL Molecular Graphics System v2.0 | Schrödinger | |

| Reagent/resource | Reference or source | Identifier or catalogue number |
|---|---|---|
| CHARMM-GUI | https://charmm-gui.org | |
| GROMACS 2022 | https://www.gromacs.org | |
| TIP3P water model | In GROMACS | |
| Nosé-Hoover thermostat | In GROMACS | |
| Parrinello-Rahman barostat | In GROMACS | |
| LINCS algorithm | In GROMACS | |
| Particle-Mesh Ewald (PME) | In GROMACS | |
| HMMER v3.1b2 (hmmsearch, hmmbuild) | http://hmmer.org | |
| UniProt ID-mapping tool | https://www.uniprot.org/id-mapping | |
| NCBI Batch Entrez | https://www.ncbi.nlm.nih.gov/sites/batchentrez | |
| Hamburger v0.2.0 | https://github.com/djw533/hamburger | |
| macsyfinder | https://github.com/gem-pasteur/macsyfinder | |
| IQ-TREE v1.6.5 | http://www.iqtree.org | |
| ggtree v1.15.6 (R package) | Bioconductor; https://bioconductor.org/packages/release/bioc/html/ggtree.html | |
| figtree v1.4.4 | http://tree.bio.ed.ac.uk/software/figtree/ | |
| ggplot2 v3.1.1 | CRAN; https://cloud.r-project.org/web/packages/ggplot2/ | |
| gggenes v0.3.2 | https://wilkox.org/gggenes/ | |
| FlowJo™ v10.4.2 | Becton Dickinson | |
| WinEDR 4.00 | John Dempster, University of Strathclyde, Glasgow, UK | |
| Clampex v10.2 | Molecular Devices | |
| UNICORN™ 6.4.1 | Cytiva | |
| BD FACSDiva | Becton Dickinson | |
| GraphPad Prism 9.5.1 | https://www.graphpad.com/ | |
| **Other** | | |
| Mini-PROTEAN TGX Precast Protein Gels (4–20%) | Bio-Rad | 4561096 |
| HiPrep 26/10 Desalting Column | Cytiva | 17508701 |
| Amicon 30,000 MWCO centrifugal filters | Fisher Scientific | 10581342 |
| Gene frame | Thermo Scientific | AB0576 |
| 1.5 thickness coverslips | VWR | CA48366-205-1 |
| HisTrap HP columns (Cytiva) | VWR | 17-5247-01 |
| HiLoad Superdex 200 16/600 column (Cytiva) | Merck | GE28-9893-35 |
| DNeasy Blood & Tissue Kit | Qiagen | 69506 |

| Reagent/resource | Reference or source | Identifier or catalogue number |
|---|---|---|
| NEBNext End Repair Module | New England Biolabs (NEB) | E6050 |
| Macherey-Nagel™ NucleoSpin™ Gel and PCR Clean-up columns | Fisher Scientific | 12303368 |
| High Sensitivity DNA ScreenTape | Agilent | 5067-5584 |
| ÄKTA pure™ 25 | Cytiva | |
| BC-525C amplifier | Warner Instruments | |
| NIDAQ-MX acquisition interface | National Instruments | |
| Tycho NT.6 system | NanoTemper | |
| Agilent 2200 TapeStation | Agilent | |
| Illumina NextSeq 2000 | Illumina | |
| QSonica Q800R Sonicator | QSonica | |
| LSRFortessa Cell Analyzer | Becton Dickinson | |

## Bacterial strains and plasmids

Bacterial strains and plasmids used in this study are detailed in Appendix Table S2. Strains of *S. marcescens*, *P. fluorescens* and *Ent. cloacae* were routinely grown at 30 °C, and *E. coli* and *B. thailandensis* at 37 °C, in liquid LB (10 g/L tryptone, 5 g/L yeast extract, 5 g/L NaCl) or on LB plates (10 g/L tryptone, 5 g/L yeast extract, 10 g/L NaCl, 1.8% select agar). Where required, growth media was supplemented with 100 μg/ml ampicillin (Amp), 100 μg/ml kanamycin (Kan), 50 μg/ml gentamicin (Gen), or 100 μg/ml streptomycin (Str). Defined chromosomal mutations, including in-frame deletions and restoration of wild-type alleles, in *S. marcescens* Db10 and *P. fluorescens* 55 were generated by allelic exchange using the plasmids pKNG101 or pMQ30, respectively (Choi and Schweizer, 2006; Murdoch et al, 2011). *S. marcescens* Db10 differs by only a single nucleotide from strain Db11 and thus the complete genome sequence of Db11 is used interchangeably for Db10 (Iguchi et al, 2014). Exogenous genes were integrated into the neutral *attB* site of *P. fluorescens* using the plasmids pJM220 (for rhamnose-inducible expression constructs) or pUC18T-miniTn7T-Gm^R P_rpsG-mScarlet (for gentamycin-resistant target strains) (Choi and Schweizer, 2006). Selected mutants of *E. coli* BW25113 were retrieved from the Keio collection (Baba et al, 2006), verified using PCR and sequencing, and if required, the kanamycin-resistance cassette was excised by transient expression of FLP recombinase from the plasmid pCP20 (Baba et al, 2006). Plasmids for arabinose-inducible gene expression were derived from pBAD18-Kn (Guzman et al, 1995) and for constitutive gene expression from pSUPROM (Jack et al, 2004). To artificially direct export of heterologously-expressed effectors to the periplasm, an OmpA signal peptide (sp; sequence MKKTAIAIAVALAGFATVA-QAAPK) was incorporated at the N-terminus of the protein. Oligonucleotide primers and details of plasmid construction are in Appendix Table S3. All bacterial strains and plasmids generated in this study are available from the corresponding author on reasonable request.

## Bacterial co-culture (competition) assays

Co-culture assays for T6SS-dependent anti-bacterial activity were based on the method originally described by (Murdoch et al, 2011). Attacker and target cells were normalised to an $OD_{600}$ of 0.5 in LB, combined at an initial ratio of 1:1 and 25 µl of the mixture was spotted onto pre-warmed LB agar plates. The co-cultures were incubated at 30 °C (*P. fluorescens*, *S. marcescens* and *Ent. cloacae* targets) or 37 °C (*E. coli* and *B. thailandensis* targets) for 4 h, and then the recovery of viable target cells was enumerated by resuspending the total population in liquid media, performing serial 10-fold dilutions, and plating on media containing the appropriate antibiotic to select the target cells.

## PEG Labelling of periplasmic/cytoplasmic cysteines

Free cysteines were labelled using methoxypolyethylene glycol maleimide (mPEG-MAL, Sigma-Aldrich). Cultures of *S. marcescens* were grown in LB at 30 °C for 5 h. Cells were collected by centrifugation at $4000 \times g$ for 10 min, washed once with HEPES/$MgCl_2$ buffer (50 mM HEPES, pH 6.8, 5 mM $MgCl_2$) and resuspended in HEPES/$MgCl_2$ buffer to a final cell density of 0.3 OD units/ml. About 80 µl of cell suspension was incubated with 5 mM mPEG-MAL and 10 mM EDTA in a final reaction volume of 100 µl HEPES/$MgCl_2$ buffer for 1 h at room temperature. Duplicate labelling reactions were performed in the presence of 1% SDS to disrupt the cell envelope and allow mPEG-MAL labelling of transmembrane and cytoplasmic cysteine residues. The labelling reaction was stopped by the addition of DTT (final concentration 100 mM), and the labelled cells were collected by centrifugation at $21,000 \times g$ for 5 min and resuspended in 100 µl of 1x SDS-PAGE loading buffer. (50 mM Tris-HCl pH 6.8, 1.6% SDS, 1.6 mM EDTA, 8% glycerol, 0.02% bromophenol blue, 1.25% β-mercaptoethanol). Samples were separated on 10% Tris-glycine gels and transferred to PVDF membrane in transfer buffer (25 mM Tris, 192 mM glycine, 15% methanol). Membranes were blocked with 5% milk powder (Marvel) in PBS + 0.1% Tween-20 and probed with a 1:1000 dilution of mouse anti-FLAG (Sigma, #F3165), with an HRP-conjugated goat anti-mouse secondary antibody (Bio-Rad #170-6516). Detection of immune-reactive bands was performed using Immobilon™ Western Chemiluminescent HRP Substrate (Millipore) and X-ray film.

## Microscopy

Strains of *S. marcescens* were grown in 25 ml minimal glucose media (40 mM $K_2HPO_4$, 15 mM $KH_2PO_4$, 0.1% $(NH_4)_2SO_4$, 0.4 mM $MgSO_4$, 0.2% w/v glucose) supplemented with 50 µM IPTG at 30 °C for 4.5 h and adjusted to an $OD_{600}$ of 0.5. Attacker and target cells were combined in an initial 3:1 ratio, and 1 µl aliquots of the mixture were spotted onto a microscope slide layered with a pad of minimal glucose medium + 50 µM IPTG solidified by the addition of 1.5% UltraPure agarose (Invitrogen) and sealed with 1.5 thickness coverslips (VWR) attached to the microscope slide with a GeneFrame (Thermo Scientific). The slides were allowed to equilibrate within the microscope chamber, pre-heated to 30 °C, for ~45 min during which time 10-15 frames containing mixed attacker-target microcolonies were selected. Imaging was performed using a DeltaVision Core widefield microscope mounted on an Olympus IX71 inverted stand with an Olympus 100×1.35 NA objective and Cascade2 EMCCD camera. Image stacks were acquired every 12 min with Z spacing of 0.3 µm. GFP (target cells) was imaged using Ex/Em 480 nm/525 nm and exposure time 100 ms, and mCherry (attacker cells) was imaged using Ex/Em 575 nm/628 nm and exposure time 150 ms. Post-acquisition, images were stored and processed using OMERO software (http://openmicroscopy.org)(Allan et al, 2012). As fluorescence intensity values are not relevant to the analysis of cell numbers and distribution, images are presented following manual adjustment across the timecourse for clarity.

## Measurement of membrane potential permeability and ROS levels using flow cytometry

Membrane potential and permeability assays were carried out using the methodology described by (Mariano et al, 2019) with minor modifications. Co-cultures between strains of *S. marcescens* were performed as described above with an initial attacker:target ratio of 1:2. Following the 4 h co-incubation, cells were collected into 1 ml of PBS, adjusted to $1 \times 10^6$ cells/ml and stained with 10 µM $DiBAC_4(3)$ and 1 µM propidium iodide for 30 min on ice in the dark. For analysis of the impact of sp-Ssp4 and sp-Ssp6 expressed heterologously in *P. fluorescens*, overnight cultures were diluted 1 in 25 in LB and incubated at 30 °C for 3 h, gene expression was induced by the addition of 0.05% L-rhamnose and the cultures were incubated for a further 2 h prior to staining with $DiBAC_4(3)$ and PI as above. Controls were prepared using exponential phase cultures of *P. fluorescens* that had been treated with 6.3 µM melittin (Sigma-Aldrich) in PBS for 2 h at 30 °C.

For detection of ROS, overnight cultures of *E. coli* MG1655 or BW25113 carrying pBAD18-Kn derived plasmids were diluted 1 in 100 into LB media containing 10 µM OxyBURST™ Green (2',7'-dichlorodihydrofluorescein diacetate, succinimidyl ester; Invitrogen #D2935) and incubated at 37 °C for 2 h. Gene expression was induced by the addition of 0.2% L-arabinose and incubation continued for a further 3 h. At each time point at or post-induction, 10 µl of induced culture was fixed with 1% formaldehyde in PBS for 30 min at 4 °C.

Samples were analysed using a LSRFortessa Cell Analyzer (BD) equipped with 488 nm and 561 nm lasers. Data analysis were performed using FlowJo v10.4.2. 20,000 and 10,000 P1 events were collected in $DiBAC_4(3)$/PI and OxyBURST experiments, respectively

## Recombinant Ssp4 production

For production of $His_6$-GST-Ssp4 and $His_6$-GST-Ssp4, *E. coli* SHuffle T7 was transformed with plasmids derived from a modified version of the commercial pGEX-6-P1 vector (Appendix Table S2). Cultures were grown to $OD_{600}$ ~0.6–0.8 in 6 L LB at 30 °C, then gene expression was induced by the addition of 0.5 mM IPTG prior to overnight growth at 16 °C. Cells were collected by centrifugation and resuspended in 150 ml of lysis buffer (50 mM Tris-HCl, pH 7.5, 0.5 M NaCl, 20 mM imidazole, 2 mM TCEP) supplemented with cOmplete EDTA-free protease inhibitor cocktail (Roche) and 50 U/ml Benzonase nuclease (Millipore). Cells were broken by pressure cell lysis at 25,000 psi and the lysate clarified by centrifugation at $48,000 \times g$ and passage through a 0.45 µm filter.

Recombinant proteins were isolated by immobilised metal affinity chromatography using a three-step isocratic elution (60 mM, 120 mM and 200 mM imidazole) and a 5 ml HisTrap HP column (Cytiva). Protein-containing fractions were pooled and incubated with 1 ml of glutathione sepharose 4B resin (Cytiva) for 1 h at room temperature. The resin was washed with 20 ml of GST buffer (20 mM Tris-HCl, pH 7.5, 0.5 M NaCl, 1 mM TCEP and 1 mM EDTA) and His-GST tagged proteins were eluted in 5 ml GST buffer containing 50 mM reduced glutathione.

To obtain untagged Ssp4, samples were exchanged into buffer containing 20 mM Tris-HCl, pH 7.5, 0.5 M NaCl, 1 mM TCEP using a HiPrep 26/10 Desalting column and incubated with 10 µl of PreScission protease (Cytiva) overnight at 4 °C. Following removal of the cleaved $His_6$-GST using a 1:1 mixture of glutathione and Ni sepharose (100 µl each, 1 h incubation at room temperature), recombinant Ssp4 was concentrated using a 30,000 MWCO centrifugal filter and further purified by size exclusion chromatography (SEC) using a HiLoad S200 16/600 column (Cytiva) in buffer containing 20 mM Tris-HCl pH 7.5, 0.5 M NaCl, 1 mM TCEP. The folded state and stability of the protein was assessed by SEC using a calibrated HiLoad S200 16/600 column and thermal calorimetry using the Tycho NT.6 system (NanoTemper). Purified Ssp4 was visualised by SDS-PAGE using 4–20% Mini-PROTEAN TGX Precast Protein Gels and followed by Instant Blue staining (Expedeon).

## Electrophysiology measurements and analysis

Planar lipid bilayers were prepared by resuspending bovine phosphatidylethanolamine lipids (Avanti Polar Lipids) in decane at a final concentration of 30 mg/mL and forming lipid bilayers across a 150-µm diameter aperture in a partition that separates two 1 mL compartments, the *cis* and the *trans* chambers (Woodier et al, 2015). KCl or $CaCl_2$ solutions at the concentrations indicated were buffered with 10 mM HEPES, pH 7.2 and added to the appropriate chamber.

Ssp4 (0.5-1 µg) was added to the *cis* chamber. The *cis* side was continuously stirred to facilitate incorporation of Ssp4 into the bilayer, and incorporation was assessed by visualisation of channel activity measured by a change in current from 0 pA. To obtain a 'mock' (no Ssp4) sample, used as a negative control, sHuffle T7 cells containing the empty pGEX expression vector, encoding only GST, were lysed and processed using the Ssp4 purification protocol. Following Ssp4 incorporation, the *trans* chamber was held at 0 mV while the *cis* chamber was clamped at different holding potentials relative to ground. The transmembrane current was measured under voltage-clamp conditions using a BC-525C amplifier (Warner Instruments). Channel recordings were low-pass filtered at 10 kHz with a four-pole Bessel filter, digitised at 100 kHz using a National Instruments acquisition interface (NIDAQ-MX, National Instruments) and recorded on a computer hard drive using WinEDR 4.00 acquisition software (John Dempster, University of Strathclyde, Glasgow, UK). Current fluctuations were measured at room temperature over 30–60 s. Recordings were filtered using a low-pass digital filter at 800 Hz (−3 dB) implemented in WinEDR 3.05.

Measurements of current amplitudes were carried out in WinEDR 3.05. The closed and main open state level was assessed manually using cursors within the analysis software. Predicted

reversal potentials were calculated using the Nernst equation. The relative $Ca^{2+}$ to $K^+$ permeability ratio ($PCa^{2+}/PX^+$) was calculated using the Fatt-Ginsborg equation(Fatt and Ginsborg, 1958).

$$PCa^{2+}/PK^+ = [K^+]/4[Ca^{2+}] \cdot \exp(E_{rev}F/RT) \cdot [\exp(E_{rev}F/RT + 1]$$

The value of $RT/F$ used in our calculations was 25.4 mV based on the recording temperature of 22 °C. The reversal potential ($E_{rev}$) was taken at the voltage where zero current was measured. Junction potentials were calculated using Clampex v10.2 (Molecular Devices) and subtracted from the reversal potential obtained for each experiment.

## Generation of saturated *P. fluorescens* 55 transposon mutant library

*E. coli* SM10λpir carrying pIT2 and *P. fluorescens* 55 were each streaked onto three LB + Amp (*E. coli*) or LB (*P. fluorescens*) plates from frozen stocks and grown overnight. Cells were recovered and resuspended to an $OD_{600}$ of 50 (*P. fluorescens*) or 100 (*E. coli*) in LB. About 100 µl of each suspension were combined and 50 µl aliquots were spotted onto LB agar, followed by incubation at 30 °C for ~6 h. Cells were resuspended in LB, diluted and plated onto ~60 LB agar plates containing 60 µg/ml tetracycline (to select for *P. fluorescens* cells containing transposon insertions) and 10 µg/ml chloramphenicol (to counterselect *E. coli*) and incubated at 30 °C for 24 h. Resulting colonies (~60,000–100,000) were resuspended in 6 ml LB to which 3 ml of 50% glycerol was added for storage at −80 °C in 100 µl aliquots.

## Co-culture and library preparation for transposon insertion site sequencing

Similar to the co-culture assays above, cells of each strain of *S. marcescens* and an aliquot of the *P. fluorescens* library were adjusted to an $OD_{600}$ of 0.5 in LB. The suspensions were combined 1:2 (attacker:target) in appropriate combinations, and $2 \times 25$ µl aliquots of the mixture were spotted onto pre-warmed LB agar plates and incubated at 30 °C for 4 h. The resulting cells were resuspended in 1 ml of PBS, and genomic DNA was extracted using the DNeasy Blood & Tissue Kit (Qiagen). Genomic DNA (~10 µg) was sheared into 250 bp fragments by sonication using the QSonica Q800R sonicator (Amplitude 25%: 20x cycles of 15 s on/off) and end-repaired using the NEBNext End Repair Module (New England Biolabs). A polyC tail was added to 1 µg of end-repaired DNA using Terminal Deoxynucleotidyl Transferase (NEB) with a mixture of 95% dCTP and 5% ddCTP as a substrate. Residual C-tailing reagents were removed using NucleoSpin™ Gel and PCR Clean-up columns (Macherey-Nagel), and DNA fragments containing transposon insertion sites were amplified by PCR using Q5 Hot Start High-Fidelity DNA Polymerase (New England Biolabs) and oligonucleotide primers as detailed in Appendix Table S3. Sample clean-up and size selection was performed using magnetic AMPure XP beads (Beckman). An initial incubation with 0.8 volumes of bead solution was used to remove long DNA fragments (>250 bp), and the fragments of interest were adsorbed from the resulting supernatant using an additional 0.4 volumes of bead solution. The beads were washed twice with 80% ethanol prior to elution of the DNA in 30 µl ultrapure $H_2O$. Index primers (NEBNext Multiplex Oligos for Illumina, New England Biolabs) were incorporated by PCR, and indexed fragments of interest were adsorbed using 1.2 volumes of AMPure XP beads. The beads were washed twice with 80% ethanol prior to elution of the DNA in

30 μl ultrapure H₂O. The quality and concentration of DNA was assessed using High Sensitivity DNA ScreenTape (Agilent) and an Agilent 2200 TapeStation.

## Generation and analysis of Tn-seq sequencing data

Prepared DNA libraries were pooled for sequencing on the Illumina NextSeq2000 instrument with a P1 reagent kit. Approximately 2.2 - 4.6 million single-end 150 bp reads were obtained per sample. Raw FASTQ files were processed for analysis using the Tn-Seq Pre-Processor (TPP) tool from the Transit package (DeJesus et al, 2015), which finds and filters the transposon sequence from each read and maps the remaining genome sequence to a reference genome (here *P. fluorescens* 55, see below) using bwa (http://arxiv.org/abs/1303.3997). Approximately 1.9–4.0 reads per sample were successfully mapped. Plots of insertions per site were generated from .wig files output by the TPP tool, and output .sam files were used in conjunction with the genome annotation to summarise counts per gene using the featureCounts algorithm of the subread software package (Liao et al, 2014).

We aimed to identify genes with significantly altered numbers of transposon-derived reads between conditions. Because the fate of individual cells in a T6SS intoxication experiment is stochastic (depending on the proximity of each prey cell to an attacker cell, for example) we found that in many cases the normalised counts per gene varied dramatically between the three biological replicates of each intoxication condition. We assume that these highly variable genes do not strongly affect fitness, and therefore the dataset for subsequent analysis was limited to genes where the coefficient of variation under each condition was <0.5. Differential expression analysis was then performed with *edgeR* version 3.38.4 (Robinson et al, 2010) to compare pairwise the three intoxication conditions ($\Delta 9$ vs $\Delta 9 + ssp4$; $\Delta 9$ vs $\Delta 9 + ssp6$; $\Delta 9 + ssp4$ vs $\Delta 9 + ssp6$) in order to identify genes significantly different between conditions (FDR <0.05). The R code for this analysis is available at https://github.com/bartongroup/MG_T6SS_tn-seq.

Separately, whole genome sequencing of *P. fluorescens* 55 was provided by MicrobesNG (https://microbesng.com), using hybrid short (Illumina) and long (Oxford Nanopore) read sequencing to generate a closed whole genome assembly with automated gene annotation.

## Structural predictions

Membrane topology predictions were generated using MEMSAT-SVM (Nugent and Jones, 2009) hosted on the PSIPred workbench (http://bioinf.cs.ucl.ac.uk/psipred/). Predictions of Ssp4 and Sip4 monomer structures were generated using AlphaFold2 (Jumper et al, 2021) hosted on Google Colab (https://colab.research.google.com). Structural predictions for various multimeric Ssp4 assemblies (putative pore structures) were generated using AlphaFold2 hosted on the University of Dundee HPC cluster. Structural representations were generated using the PyMOL Molecular Graphics System, Version 2.0 (Schrödinger).

## Molecular dynamics simulations

The structural models of the Ssp4₁₁₄₋₃₀₂ oligomers and monomer were embedded into 1-palmitoyl-2-oleoyl-*sn*-glycerol-3-phosphatidyl ethanolamine (POPE) membranes and aligned in the membrane along the subunit principal axis using the CHARMM-GUI server (Jo et al, 2008). The initial box sizes measured ~125 Å in z-dimension and varied between 90 × 90 Å and 120 × 120 Å in x,y-dimension. The CHARMM36m force field was used for the protein, lipids and ions (KCl at a concentration of 0.15 M), and the TIP3P water model was used to model water molecules (Huang et al, 2017; Jorgensen et al, 1983). All molecular dynamics (MD) simulations were carried out with the GROMACS 2022 software package (Van der Spoel et al, 2005). Energy minimisation and equilibration was performed according to the protocols provided by the CHARMM-GUI server (Jo et al, 2008). A further equilibration step comprised unbiased simulations of 100 ns length without restraints or an electric field, using a 2 fs integration timestep. Equilibration was followed by three-fold replicated production simulations under a membrane voltage of at least 250 ns length each. For the monomer and tetramer, simulations were extended by an additional length of 1 μs. The production simulations were conducted in the NPT ensemble, with the temperature controlled at 310 K using the Nosé-Hoover thermostat and the pressure semi-isotropically maintained at 1 bar using the Parrinello-Rahman barostat (Evans and Holian, 1985; Parrinello and Rahman, 1981). All production simulations used a 2 fs integration timestep. To constrain bond lengths involving H atoms, the LINCS algorithm was employed; long-range electrostatic interactions were modelled using the Particle-Mesh Ewald method (Darden et al, 1993; Hess et al, 1997). Membrane voltages were generated using an applied external electric field (Aksimentiev and Schulten, 2005).

## Identification of Ssp4-like effector family

Distribution of Ssp4-like effectors across known protein space was determined by searching the UniProt database (UniProt, 2023) for Ssp4-like proteins, using hmmsearch from the HMMER suite v3.1b2 (Eddy, 2011). An 'Ssp4' model was constructed, using hmmbuild, from a small, manually-curated alignment of non-redundant ssp4 homologues identified using BLASTp (Camacho et al, 2009). After initial analysis of all hits and their taxonomic distribution, along with visual analysis of protein alignments, hits with an HMMER score >100 were considered to be homologues. No length coverage cutoff was applied in order to include any protein fragments in the Uniprot database and therefore span as broad a taxonomic space as possible, however 95% of the hits had coverage of >200 amino acids and including regions corresponding to the predicted transmembrane helices of Ssp4 (Dataset EV2; Appendix Fig. S7). Accessions for publicly available genomes or contigs corresponding with Ssp4-like UniProt IDs identified using this approach were located using the ID-mapping tool in UniProt (https://www.uniprot.org/id-mapping), matching the UniProt IDs against the target database EMBL/GenBank/DDBJ. Annotated genomes corresponding to these accessions were then downloaded using the NCBI batch Entrez tool (https://www.ncbi.nlm.nih.gov/sites/batchentrez). Genomic regions of ~20 kb containing the gene encoding the Ssp4-like protein (10 kb upstream and downstream of the Ssp4-encoding gene) were then retrieved from these genomes using hamburger (Mariano et al, 2019; Williams, 2022). T6SS genes were also identified during the search, using models within hamburger (Mariano et al, 2019; Williams, 2022). T3SS genes were identified by hmmsearch (Eddy, 2011) using models from macsyfinder (Abby et al, 2016). Several potential Ssp4 hits with a low hmmsearch sequence score matched the Type III Secretion System (T3SS) translocon protein SipB, suggesting that they were

false positives. Subsequently, all hmmsearch hits with a sequence score of less than 100 were removed from further analysis. Trees of Ssp4-like protein sequences were drawn using IQ-TREE (v1.6.5) (Nguyen et al, 2014) with 1000 ultrafast bootstraps (Hoang et al, 2018) using models chosen by modelfinder (Kalyaanamoorthy et al, 2017). Trees were drawn from alignments created by hmmsearch(-Eddy, 2011). Trees were visualised using the R packages ggtree (v1.15.6) (Yu et al, 2017) and figtree (v1.4.4) (http://tree.bio.ed.ac.uk/software/figtree/), and associated genomic context depicted using ggplot2 (v3.1.1) (Wickham, 2016) and gggenes (v0.3.2) (https://wilkox.org/gggenes/).

### Statistical analysis and experimental design

Unless stated otherwise, statistical analysis was performed using GraphPad Prism 9; significant differences between independent biological replicates were determined using one-way ANOVA with Tukey's or Dunnett's post-tests and data were tested for normal distribution using the Shapiro–Wilks test. Sample sizes (number of biological replicates) were chosen according to normal practice in the field and experimental feasibility. No blinding was performed.

## Data availability

The complete genome sequence of *P. fluorescens* 55 and the Tn-seq sequencing data are publicly available in the NCBI Genome (accession CP179688) and GEO (accession GSE301252) databases, respectively. The R code for the differential expression analysis is available at https://github.com/bartongroup/MG_T6SS_tn-seq. All other data supporting the findings of this study are available within the paper and its supplementary material.

The source data of this paper are collected in the following database record: biostudies:S-SCDT-10_1038-S44318-025-00587-x.

## Peer review information

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

## Acknowledgements

This work was supported by Wellcome (grant numbers 104556/Z/14/Z, Senior Research Fellowship SJC; and 220321/Z/20/Z, Senior Research Fellowship Renewal SJC), UKRI Medical Research Council (grant numbers MR/K000111X/1, New Investigator Research Grant, SJC; and MR/T041811/1, Future Leaders Fellowship, MB), UKRI Biotechnology and Biological Sciences Research Council (grant numbers BB/M010996/1 and BB/T00875X/1, EASTBio PhD studentships KM and ATS), Academy of Medical Sciences (grant number SBF005/1096, Springboard Grant, MB; the Springboard scheme is funded by Wellcome, the Department of Business, Energy, and Industrial Strategy UK, the British Heart Foundation, and Diabetes UK) and the British Heart Foundation (grant number FS/PhD/22/2932, non-clinical PhD studentship QWH). We thank Yi-Chia Liu and Giuseppina Mariano for the generation of strains; Daan van Aalten for the gift of pHis-GEX-6P-1; Laura Monlezun for establishing the purification of GST-fused Ssp4; Pete Hedley for assistance with Tn-seq sequencing; and the Flow Cytometry and Cell Sorting Facility, High Performance Computing Facility and Dundee Imaging Facility at the University of Dundee for access to facilities and expert assistance. For the purpose of Open Access, the authors have applied a CC BY public copyright licence to any Author Accepted Manuscript version arising from this submission.

## Author contributions

**Mark Reglinski**: Conceptualisation; Formal analysis; Investigation; Visualisation; Writing—original draft. **Quenton W Hurst**: Formal analysis; Investigation. **David J Williams**: Formal analysis; Investigation; Visualisation; Writing—original draft. **Marek Gierlinski**: Formal analysis. **Alp Tegin Şahin**: Investigation. **Katharine Mathers**: Investigation. **Adam Ostrowski**: Investigation. **Megan Bergkessel**: Formal analysis; Investigation; Visualisation; Writing—original draft. **Ulrich Zachariae**: Formal analysis; Supervision; Investigation; Visualisation; Writing—review and editing. **Samantha J Pitt**: Formal analysis; Supervision; Visualisation; Writing—original draft; Writing—review and editing. **Sarah J Coulthurst**: Conceptualisation; Formal analysis; Supervision; Funding acquisition; Visualisation; Writing—original draft; Writing—review and editing.

Source data underlying figure panels in this paper may have individual authorship assigned. Where available, figure panel/source data authorship is listed in the following database record: biostudies:S-SCDT-10_1038-S44318-025-00587-x.

## Disclosure and competing interests statement

The authors declare no competing interests.

# Expanded View Figures

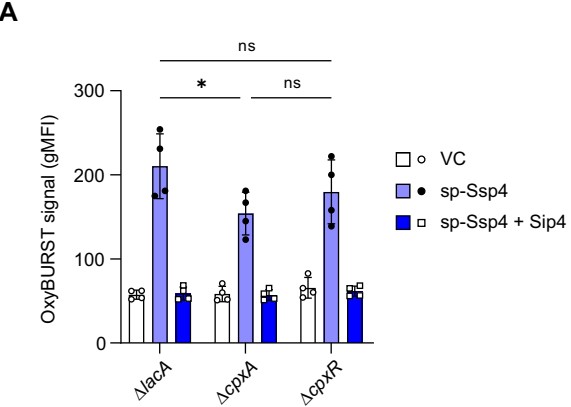

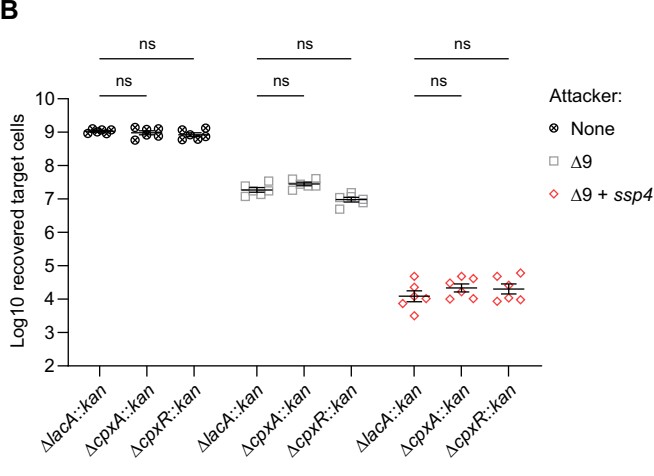

**Figure EV1. Loss of the Cpx system in *E. coli* has little or no impact on susceptibility to Ssp4 intoxication.**

(A) Quantification of OxyBURST Green intensity from strains of *E. coli* BW25113 (ΔlacA, ΔcpxA, ΔcpxR) carrying plasmids directing expression of sp-Ssp4 or sp-Ssp6, following 3 h induction. (B) Recovery of *E. coli* BW25112 carrying control (ΔlacA::kan) or Cpx (ΔcpxA::kan, ΔcpxR::kan) gene deletions, following co-culture with attacking strains of *S. marcescens* Db10 lacking known anti-bacterial effectors (Δ9) or delivering only Ssp4 (Δ9 + ssp4). None, no-attacker; kan, kanamycin-resistance gene replacing the deleted gene and providing selection for target cells. Data were presented as mean ± SEM with individual data points overlaid (n = 4 or n = 6 biological replicates in panels (**A**, **B**), respectively); *P < 0.05, ns not significant, one-way ANOVA with Tukey's test; for clarity, only selected comparisons are displayed. P values from left to right (**A**) P = 0.0189, P = 0.5232, P = 0.7222; (**B**) P > 0.9999, P = 0.9978, P = 0.9339, P = 0.4848, P = 0.6823, P = 0.8057. Source data are available online for this figure.

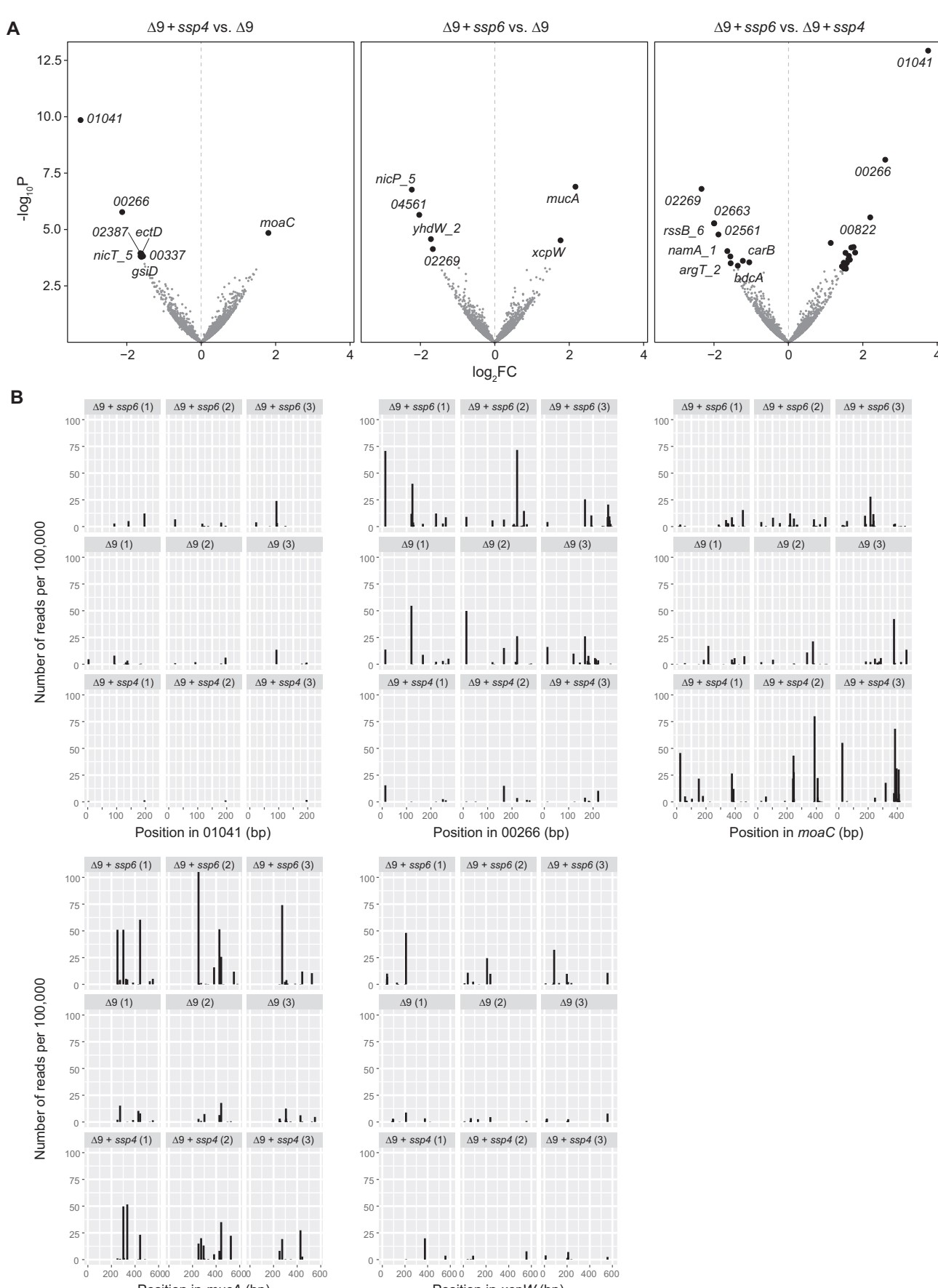

◀ **Figure EV2.  All three Tn-seq pairwise comparisons and individual insertion sites in genes of interest.**

(A) Volcano plots summarising the change in recovery of *P. fluorescens* 55 transposon insertion mutants between control (Δ9) and Ssp4-delivering attackers (left), between control and Ssp6-delivering attackers (middle), and between Ssp4- and Ssp6-delivering attackers (right) on a per gene basis. Log2 fold change in normalised read count is plotted against —log10 *P* value and genes significantly altered between condition (FDR <0.05, EdgeR's quasi-likelihood *F*-test (QLF test) with Benjamini–Hochberg correction) are highlighted as black dots, with all (left, middle) or selected (right) gene annotations. (B) Position and number of sequencing reads for individual transposon insertion sites across five genes of interest in each replicate for each attacking strain.

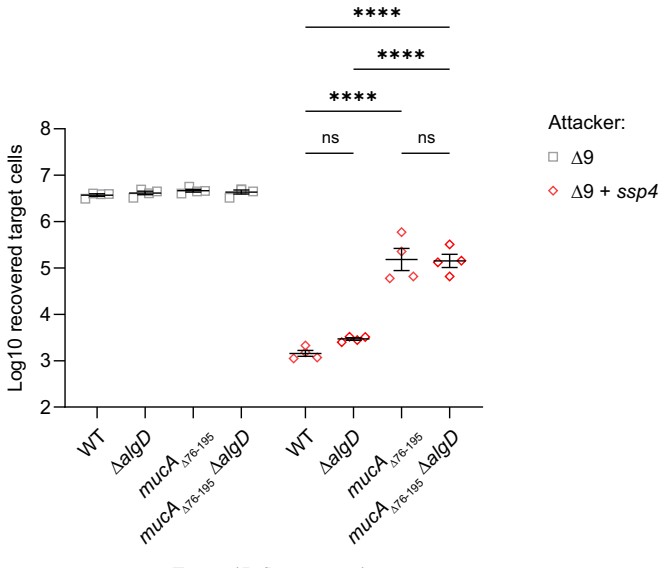

**Figure EV3.   Deletion of an alginate biosynthesis gene does not affect the resistance of the *P. fluorescens mucA*<sub>Δ76–195</sub> mutant to T6SS attacks.**

Recovery of wild type (WT) or defined deletion mutants (Δ*algD*, *mucA*<sub>Δ76–195</sub>, or *mucA*<sub>Δ76–195</sub> Δ*algD*) of *P. fluorescens* 55, following co-culture with attacking strains of *S. marcescens* Db10 as indicated. Data are presented as mean ± SEM with individual data points overlaid (*n* = 4 biological replicates); ****$P < 0.0001$, ns not significant; one-way ANOVA with Tukey's test; for clarity, only selected comparisons are displayed. *P* values from left to right $P = 0.4335$, $P < 0.0001$, $P < 0.0001$, $P < 0.0001$ and $P > 0.9999$. Source data are available online for this figure.

