## [Peer Review File · The EMBO Journal]

A widely-occurring family of pore-forming effectors broadens the impact of the Serratia Type VI secretion system

Mark Reglinski, Quenton Hurst, David Williams, Marek Gierliński, Alp Sahin, Katharine Mathers, Adam Ostrowski, Megan Bergkessel, Ulrich Zachariae, Samantha Pitt, and Sarah Coulthurst

Corresponding author: Sarah Coulthurst (s.j.coulthurst@dundee.ac.uk)

Review Timeline:

Submission Date:	4th Feb 25
Editorial Decision:	26th Feb 25
Revision Received:	22nd Jul 25
Editorial Decision:	19th Aug 25
Revision Received:	9th Sep 25
Accepted:	23rd Sep 25

Editor: Ieva Gailite

Transaction Report:

Dear Dr. Coulthurst,

Thank you for submitting your manuscript for consideration by the EMBO Journal. We have now received comments from a full set of reviewers, which are included below for your information.

As you will see, all reviewers are generally positive in their assessment and appreciate the contribution of the study to the research field. At the same time, they indicate a number of concerns that would be important to address in the revised study. From my side, I find these points generally reasonable. Therefore, I invite you to address these comments in a revised manuscript. I think that it would be useful to discuss the revision in more detail via email or phone/videoconferencing - please let me know which option you prefer.

We generally allow three months as standard revision time, which can be extended to six months in the case of major revisions. Should you foresee a problem in meeting this deadline, please let us know in advance to discuss an extension. As a matter of policy, competing manuscripts published during this period will not negatively impact on our assessment of the conceptual advance presented by your study. However, please contact me as soon as possible upon publication of any related work to discuss the appropriate course of action.

When preparing your letter of response to the referees' comments, please bear in mind that this will form part of the Review Process File and will therefore be available online to the community. For more details on our Transparent Editorial Process, please visit our website: <https://www.embopress.org/page/journal/14602075/authorguide#transparentprocess>. Please also see the attached instructions for further guidelines on preparation of the revised manuscript.

Please feel free to contact me if have any further questions regarding the revision. Thank you for the opportunity to consider your work for publication, and I look forward to discussing your revision with you.

With best wishes,

Ieva

We realize that it is difficult to revise to a specific deadline. In the interest of protecting the conceptual advance provided by the work, we recommend a revision within 3 months (27th May 2025). Please discuss the revision progress ahead of this time with the editor if you require more time to complete the revisions.

Referee #1:

The study of Ssp4, a putative pore-forming toxin delivered by the Type VI Secretion System (T6SS), holds significant potential for advancing our understanding of bacterial pathogenicity and interbacterial competition. Pore-forming toxins play a critical role in modulating cellular environments, often disrupting membrane integrity to facilitate bacterial survival and proliferation. Understanding the biophysical properties of these toxins, particularly their ion selectivity and conductance characteristics, is essential for elucidating their mechanisms of action. The authors' efforts to characterize the electrophysiological behavior of Ssp4, coupled with structural modeling and molecular dynamics simulations, represent a valuable contribution to this field. However, to fully realize the impact of these findings, it is imperative that the experimental methodologies and data interpretations meet rigorous scientific standards.

Electrophysiology Section Review:

The electrophysiological characterization of Ssp4 presents several areas that require improvement. The authors are encouraged to address these issues to strengthen the validity of their conclusions.

The reported conductance measurements are based on an $n = 4$, though it is unclear what this sample size represents. If n refers to the number of observed jump events, this is insufficient, as even the provided traces suggest a higher number of jump events. Alternatively, if n corresponds to four independent traces, with histograms generated for each to differentiate between open and closed states, then the authors should provide these individual histograms. This would allow readers to assess the consistency and robustness of the data. Moreover, clarification is needed on how these histograms were combined to yield the final conductance values, as the reported standard deviation appears unusually low, raising concerns about the statistical representativeness of the data.

The single-channel conductance value of 18.4 pS at 500 mM KCl is comparable to that of gramicidin A under similar conditions [1]. Therefore, we could extrapolate that Ssp4 pore size should be around ~ 0.4 nm in radius [2], further supporting the ion-selective nature of Ssp4 pores. But, a major limitation in the current study is the absence of control traces from protein-free bilayers. Without such controls, it is impossible to exclude the possibility that the observed noise originates from intrinsic membrane dynamics rather than discrete open and closed states attributable to Ssp4.

The reversal potential data similarly lack statistical support, and without more rigorous statistical treatment, these findings remain tentative. The standard deviation is again surprisingly small. In any case, the measured reversal potential of -14.8 mV could be approximated to an ion permeability ratio $\sim 9-10$. This value would indicate "a preference for cations over anions", as it is correctly interpreted, but does not indicate that Ssp4 forms cation-selective pores as suggested in other parts of the manuscript. This is an essential distinction because weakly selective channels usually have a K^+/Cl^- permeability ratio (P_{K^+}/P_{Cl^-}) < 5 , while ideal cation-selective channels can reach values of 100 to 1000 [3,4], which means that cation-selective channels are highly specific and tightly regulate the ionic transport across the membranes, while weakly selective channels, like Ssp4, preferentially transport cations but without totally excluding anions.

The calcium experiments introduce additional complications. The conditions under which KCl and $CaCl_2$ are compared are problematic, as equimolar concentrations of these salts do not equate to equivalent ionic conditions—specifically, the chloride concentration is doubled in the presence of $CaCl_2$. This discrepancy complicates the interpretation of the results. Furthermore, the linear current-voltage relationship reported at very low current levels ($I < 1$ pA) with only three replicates ($n = 3$) lacks

sufficient statistical significance, making it difficult to draw reliable conclusions. Also, on Fig. 4d lower panel (210 mM KCl cis, 210 mM CaCl₂ trans), the current appears voltage-independent at positive voltages, and the linear extrapolation applied by the authors is questionable.

Additionally, the author's interpretation of the calcium experiments appears oversimplified. They attribute the observed effects solely to a potential Ca²⁺-dependent conformational change in the Ssp4 pore, without considering the well-documented influence of Ca²⁺ on phosphatidylethanolamine (PE) membranes [5,6]. Ca²⁺ ions interact with membrane lipids, affecting properties such as membrane fluidity, tension, and potential. These interactions can induce tighter lipid packing, alter local electric fields, and affect bilayer asymmetry—all of which could influence ion flow and confuse the attribution of observed effects to the Ssp4 pore itself. To clarify this, the authors should include control experiments with PE membranes exposed to Ca²⁺ in the absence of Ssp4, varying Ca²⁺ concentrations and consider testing other lipid compositions to assess the consistency of their observations. Alternatively, they could exclude these results from the manuscript given that they are unnecessary to support the central message of the manuscript, which is that Ssp4 assembles ion-selective pores. Finally, the study seems limited to a single lipid composition (PE), as indicated in the methods. Exploring additional lipid mixtures would provide valuable insights into how the lipid environment influences pore formation and function.

1. Queralt-Martín M, López ML, Aguilera-Arzo M, Aguilera VM, Alcaraz A: Scaling Behavior of Ionic Transport in Membrane Nanochannels. *Nano Lett* 2018, 18:6604-6610.
2. Andersen OS, Koeppe II RE, Roux B: Gramicidin Channels: Versatile Tools. In *Biological Membrane Ion Channels*. . Springer New York; [date unknown]:33-80.
3. Hille B: Ionic channels of Excitable Membranes 2nd edition. *J Electrochem Soc* 1987, 134.
4. Aguilera VM, Queralt-Martín M, Aguilera-Arzo M, Alcaraz A: Insights on the permeability of wide protein channels: measurement and interpretation of ion selectivity. *Integr Biol* 2011, 3:159-172.
5. Melcrová A, Pokorna S, Pullanchery S, Kohagen M, Jurkiewicz P, Hof M, Jungwirth P, Cremer PS, Cwiklik L: The complex nature of calcium cation interactions with phospholipid bilayers. *Sci Rep* 2016, 6:38035.
6. Marra J, Israelachvili J: Direct measurements of forces between phosphatidylcholine and phosphatidylethanolamine bilayers in aqueous electrolyte solutions. *Biochemistry* 1985, 24:4608-4618.

Review Structural modelling and molecular dynamics simulation:

As pointed out by the authors, "Upon delivery to their target cellular compartment, α -PFT monomers typically undergo a conformational change resulting in exposure of hydrophobic or amphipathic helices, followed by oligomerisation and concomitant membrane insertion to form the mature pore³⁴. Such toxins are thus well suited to delivery by the T6SS, where two forms of the protein are likely to exist: a pre-secretion form which is soluble and compact to allow loading into the Hcp tube, and an active form with exposed hydrophobic helices primed for membrane insertion."

Current limitations of AlphaFold (AF) for predicting the structure of pore-forming toxins include:

AF struggles with accurately predicting the membrane insertion steps of pore-forming toxins, as these involve dynamic conformational changes that are not easily captured by static structural predictions.

Furthermore, predicting oligomeric structures, such as tetramers or larger assemblies, is particularly difficult. The dynamic process of oligomerization, crucial for pore formation, is challenging to model accurately with current AF capabilities.

The influence of the lipid environment on toxin structure and function is challenging to integrate into AF predictions. Factors such as lipid composition and membrane dynamics are critical for pore formation but are not accounted for in AF's modeling process.

The accuracy of AF predictions relies heavily on the availability of diverse and extensive Multiple Sequence Alignments (MSAs). For many pore-forming toxins, especially those without well-characterized homologs, this can be a significant limitation.

How do the authors interpret AF results? What conformation is AF predicting for the Ssp4 monomer, the soluble, or the membrane form with exposed hydrophobic helices?

Do they observe significant conformational differences between the AF-predicted monomeric and tetrameric structures?

What are the predicted aligned error (PAE) scores for the tetramer? What are the scores for the regions involved in protein-protein interactions?

Do they observe significant conformational changes in Ssp4 following the 1-microsecond MD simulations of the AF tetramer in a POPC model membrane?

In general, bacterial membranes are dominated by phosphatidylethanolamine (PE), phosphatidylglycerol (PG), and cardiolipin

(CL). So, why did they choose to carry out MD simulations on a POPC model membrane?

Have they attempted to validate experimentally that Ssp4 assembles tetrameric pores, for example, by analytic ultracentrifugation or native Mass Spectrometry of Ssp4 pores embedded in detergent micelles?

In the legend of Fig. 5c, the authors present an overlay of K⁺ ion positions from a 1 μ s simulation, suggesting a preferred ion permeation pathway across the pore formed by Ssp4. However, in the discussion section, the authors state: "Our modeling suggests that Ssp4 forms tetrameric pores with water-filled, ion-conducting channels..." In their MD simulations, do they observe ionic transport through several channels within the Ssp4 tetramer? What are the diameters of these channels? Do these diameters correlate with those estimated by their electrophysiology measurements and those of gramicidin A?

What residues line the lumen of the Ssp4 tetrameric channel? To validate their model, they could consider mutating some of these residues and measuring the effect on ion conductance.

Considering the limitations of AF for predicting the structures of PFTs and the lack of experimental validation of the AF model, their computational conclusions need to be reevaluated. For example, I would strongly disagree with the following statement: "Here, we present the first high confidence structural prediction for such a pore". What makes them think they obtained a high-confidence structural prediction of Ssp4 pore?

Referee #2:

In this manuscript, the authors show that T6SS toxins Ssp4 and Ssp6 are pore-forming toxins with wide range and narrow range of targets, respectively. Ssp4 acts from the periplasm, kills target cells without their lysis and has a transmembrane topology. The authors show that Ssp4 depolarizes the membrane by forming pores (MD simulation and measurements) and that this also leads to ROS production. The authors further performed a Tn-seq experiment to identify genes required for resistance and sensitivity to Ssp4 in *P. fluorescens*. This and subsequent mutagenesis identified that deletion of C-term of mucA provides protection against Ssp4 but also other toxins. Deletion of moaC provided a relatively modest protection as well. This is a manuscript with high-quality data and figures as well as clear explanations. The experiments are logical, with sufficient replicates, and well-controlled as well as properly analyzed and interpreted.

Comments:

It would be interesting to delete more genes in the same pathway as moaC to test if the observed phenotype is dependent on moaC in particular or rather on the whole pathway.

For the mucA truncation mutant, it would make sense to test if an additional deletion of a gene in the EPS synthesis pathway would restore the toxin sensitive phenotype.

This is not essential, but could you try to identify more resistant mutants by expressing Ssp4 in the periplasm of target cells rather than by mixing them with the T6SS⁺ attacker?

While you clearly show that Ssp4 targets bacterial membrane, is there any evidence that it fails to target eukaryotic membranes?

Referee #3:

In this manuscript, Reglinski et al. perform a comprehensive analysis of the *Serratia* T6SS effector Ssp4. They use various genetic, biochemical, and biophysical techniques to investigate its toxicity, target range, mechanism of action, and distribution. Furthermore, they use a Tn-seq approach to identify a mechanism that enables target protection against Ssp4 intoxication. Overall, this is an interesting, well-written paper. The flow and rationale are clear, and most conclusions are well supported by the results. I have a few comments for the authors to consider.

Major comments:

1. The authors suggest that Ssp4 and Ssp6 are investigated "in isolation" when expressed in a D9 strain (lines 117-119). However, there appears to be a significant difference in intoxicating either *E. coli* or *P. fluorescens* targets when comparing the tssE mutant and D9 attacker strains. This suggests that additional effectors are delivered by this T6SS. Therefore, the conclusions could be toned down and discussed in the light of a possible synergistic effect of Ssp4 or Ssp6 with other effectors, since it is possible that they are not truly examined individually in the D9 strain.
2. lines 123-125: this is an over-reaching conclusion to base on only 2 target strains. On the same topic - Lines 307-308 and 310-312: To draw this conclusion and make these claims, the author should test more than 2 target strains.
3. Fig. 3b: ~7% of the cells being affected seems like a very low percentage, especially when the effector is expected to be over-expressed in all these cells. Can the authors comment on that? Also, did the authors attempt to use a TAT signal to send the effectors to the periplasm? Is it possible that Ssp6 is not toxic when ectopically expressed in *P. fluorescens* because it is not properly transported and folded in the periplasm when it passes unfolded through the Sec system?

4. The section on structure and fold analyses of Ssp4 (starting at line 174) is rather weak and speculative in my view. The MD analysis is based on a biased assumption that Ssp4 functions as a tetramer, which itself is based on a predicted structure that is not similar to any known and proven structure. I think that without further experimental data supporting a tetrameric complex, this section is too speculative. If the authors choose to keep it as part of the work, I suggest moving it to the supplementary materials and discussing the biases and caveats more thoroughly.

5. MucA was identified in the screen as protecting the target cells from Ssp6. It is unclear:

A) why would this happen if Ssp6 isn't toxic to this target to begin with.

B) Why was this "protection" not observed in the competition assay in Suppl. Fig. 8?

C) Did the authors consider a polar effect of the Tn mutation?

Minor comments:

1. Line 131: it is unclear whether these are 15-20% of the target cell population or the entire population (attackers+targets).

2. Fig. 3c: Ssp6 appears to associate with the membrane without being toxic. Will every protein that is sent to the periplasm using a signal-peptide be associated with the membrane fraction in this assay? What I mean is that perhaps membrane association is an artifact in this case?

3. Can the authors please explain why the ectopic expression of Ssp4 and evaluation of its effect on membrane depolarization were done in *P. fluorescens*, whereas the ROS assays are performed by expressing Ssp4 in *E. coli* cells?

4. Fig. 6: The effect of gentamicin, while possibly passing the specific statistical test at the 3-h time point, is minor. I am not sure how it adds to the story. Furthermore, the effect of Ssp6 in Fig. 6c appears comparable to that of gentamicin at this time point. Did the authors test for statistical significance between the 0 and 3-h time points in the Ssp6 samples?

5. Lines 217-227: This section leads to more questions than answers. I don't think it contributes much to the manuscript, and the authors might consider removing it.

6. Line 239: is the term "Ssp6 intoxication" appropriate in light of the observation that Ssp6 is not toxic to *P. fluorescent*?

7. Line 244: Can the authors please clarify why they chose these 2 genes and not others?

8. A list of the identified Ssp4 homologs should be provided as supplementary material. It would also be informative to determine whether the homology includes a conserved motif or structural features, such as the same number of TM helices. Please also specify in the Methods section what was the coverage cutoff when deciding what is a likely Ssp4 homolog, and whether it always included the TM region of the proteins?

9. Are the predicted Ssp4 homologs immunity proteins similar, or are there different families?

10. Lines 305-307: Similar observations were made previously with other T6SS effectors. Perhaps it is appropriate to discuss these as well?

11. It would be nice to see that MucA complementation re-sensitizes the target cell to Ssp4 intoxication.

12. Lines 404-406: Do the *mucA* mutants form mucoid colonies?

Response to Reviewers' Comments (EMBOJ-2025-120361)

We would like to thank the Reviewers for taking the time to positively and constructively review our manuscript. The manuscript has been revised in response to their comments and suggestions and has been significantly improved as a result. The Reviewers' comments have been reproduced (in black text) and our responses are included below each point (in blue text).

Referee #1:

The study of Ssp4, a putative pore-forming toxin delivered by the Type VI Secretion System (T6SS), holds significant potential for advancing our understanding of bacterial pathogenicity and interbacterial competition. Pore-forming toxins play a critical role in modulating cellular environments, often disrupting membrane integrity to facilitate bacterial survival and proliferation. Understanding the biophysical properties of these toxins, particularly their ion selectivity and conductance characteristics, is essential for elucidating their mechanisms of action. The authors' efforts to characterize the electrophysiological behavior of Ssp4, coupled with structural modeling and molecular dynamics simulations, represent a valuable contribution to this field. However, to fully realize the impact of these findings, it is imperative that the experimental methodologies and data interpretations meet rigorous scientific standards.

We thank the Reviewer for appreciating the potential of our work and taking the time to provide extensive comments on the manuscript.

Electrophysiology Section Review:

The electrophysiological characterization of Ssp4 presents several areas that require improvement. The authors are encouraged to address these issues to strengthen the validity of their conclusions.

The reported conductance measurements are based on an $n = 4$, though it is unclear what this sample size represents. If n refers to the number of observed jump events, this is insufficient, as even the provided traces suggest a higher number of jump events. Alternatively, if n corresponds to four independent traces, with histograms generated for each to differentiate between open and closed states, then the authors should provide these individual histograms. This would allow readers to assess the consistency and robustness of the data. Moreover, clarification is needed on how these histograms were combined to yield the final conductance values, as the reported standard deviation appears unusually low, raising concerns about the statistical representativeness of the data.

We apologise that the wording for this section was unclear. $N=4$ refers to the number of recordings, not the number of observed jump events; we have reworded the methods to make this clear. The method of manual fitting within single channel analysis software is an accepted method (doi: 10.1523/JNEUROSCI.3890-08.2008, doi: 10.1074/jbc.M115.661280), and is more appropriate where recordings have multiple channels incorporated into the bilayer or have low frequency openings. As suggested by the Reviewer, we also calculated the main peaks using all-point amplitude histograms fitted with weighted Gaussian distributions and used these values to construct IV relationships. As expected, the results using this method are consistent with our original analysis (see Figure R1, below). As highlighted above, where recordings had multiple channels gating in the bilayer or

channels displayed low frequency openings, the open state was sometimes masked in the histogram but was resolvable using manual fitting. For this reason, we have decided to keep with our original method of analysis as we feel that this is more accurate.

Figure R1. Ssp4 current-voltage relationships constructed using all points amplitude histograms fitted with weighted Gaussian distributions (pink) and manual current fitting within the analysis software (black). Data are displayed as mean \pm SD from 3 or 4 independent experimental recordings

The single-channel conductance value of 18.4 pS at 500 mM KCl is comparable to that of gramicidin A under similar conditions [1]. Therefore, we could extrapolate that Ssp4 pore size should be around \sim 0.4 nm in radius [2], further supporting the ion-selective nature of Ssp4 pores. But, a major limitation in the current study is the absence of control traces from protein-free bilayers. Without such controls, it is impossible to exclude the possibility that the observed noise originates from intrinsic membrane dynamics rather than discrete open and closed states attributable to Ssp4.

We thank the reviewer for highlighting this. We have now included all relevant controls. To obtain a 'mock' (no Ssp4 protein) sample, sHuffle T7 cells containing the empty pGEX expression vector were lysed and processed using the Ssp4 purification protocol. The resulting mock sample was used as a negative control. Incorporation of this mock Ssp4-free preparation into bilayers under identical conditions as the recombinant purified Ssp4 showed no functional pore activity. We have now included this data in Appendix Figure S4 and the information in the Methods and Results sections.

The reversal potential data similarly lack statistical support, and without more rigorous statistical treatment, these findings remain tentative. The standard deviation is again surprisingly small. In any case, the measured reversal potential of -14.8mV could be approximated to an ion permeability ratio $\sim 9-10$. This value would indicate "a preference for cations over anions", as it is correctly interpreted, but does not indicate that Ssp4 forms cation-selective pores as suggested in other parts of the manuscript. This is an essential distinction because weakly selective channels usually have a K^+/Cl^- permeability ratio (P_{K^+}/P_{Cl^-}) < 5 , while ideal cation-selective channels can reach values of 100 to 1000 [3,4], which means that cation-selective channels are highly specific and tightly regulate the ionic transport across the membranes, while weakly selective channels, like Ssp4, preferentially transport cations but without totally excluding anions.

The SD is small because the current always reverses at the same potential and the experimental data is consistent across replicates. This is what you would expect for a regulated pore displaying a consistent selectivity profile. It would be more surprising if the SD values were larger, which would suggest heterogeneity in the data, or a more dysregulated pore architecture.

We agree with the reviewer that the pore shows a preference rather than is 'selective' for cations and have changed the text accordingly throughout the manuscript.

However, it should be highlighted that permeability would also very much depend on ion availability. For example, the cardiac Ca^{2+} channel RyR2 shows little selectivity for Ca^{2+} over K^+ , but due to the high Ca^{2+} gradient across the SR, it behaves as a Ca^{2+} channel. Which ion Ssp4 has the ability to conduct will depend on the ionic gradient(s) dissipated and will vary with the extracellular environment of the intoxicated cell. We have highlighted this in the discussion as it is an important consideration.

The calcium experiments introduce additional complications. The conditions under which KCl and $CaCl_2$ are compared are problematic, as equimolar concentrations of these salts do not equate to equivalent ionic conditions—specifically, the chloride concentration is doubled in the presence of $CaCl_2$. This discrepancy complicates the interpretation of the results.

We thank the reviewer for their careful consideration of the data. We have now estimated the Ca^{2+}/K^+ permeability ratio using the Fatt-Ginsborg equation which allows us to calculate the relative divalent to monovalent permeability ratio. This is detailed in the Methods section. We have discussed the importance of the Cl^- component in the Discussion. We agree with the reviewer that the ability of Ssp4 to conduct anions as well as cations may contribute to its greater toxicity in vivo.

Furthermore, the linear current-voltage relationship reported at very low current levels ($I < 1$ pA) with only three replicates ($n = 3$) lacks sufficient statistical significance, making it difficult to draw reliable conclusions. Also, on Fig. 4d lower panel (210 mM KCl cis, 210 mM $CaCl_2$ trans), the current appears voltage-independent at positive voltages, and the linear extrapolation applied by the authors is questionable.

We agree that this is a very interesting preliminary finding that requires further investigation. For Figure 4D (lower panel) we have now removed the use of a linear regression fit to obtain an E_{rev} value, and have discussed permeability only with respect to the upper panel, where we can accurately measure the E_{rev} . For calculation of the conductance value for Fig 4D (lower panel), we have only used the negative values where the current-voltage relationship is ohmic. We would like to

highlight that N=3 is the number of experimental replicates not the number of open state levels measured within each recording. We have reworded the methods section to make this clear. It is also important to note that the interpretations that we previously described from the data shown in Fig 4D remain unaltered and consistent.

Additionally, the author's interpretation of the calcium experiments appears oversimplified. They attribute the observed effects solely to a potential Ca^{2+} -dependent conformational change in the Ssp4 pore, without considering the well-documented influence of Ca^{2+} on phosphatidylethanolamine (PE) membranes [5,6]. Ca^{2+} ions interact with membrane lipids, affecting properties such as membrane fluidity, tension, and potential. These interactions can induce tighter lipid packing, alter local electric fields, and affect bilayer asymmetry—all of which could influence ion flow and confuse the attribution of observed effects to the Ssp4 pore itself. To clarify this, the authors should include control experiments with PE membranes exposed to Ca^{2+} in the absence of Ssp4, varying Ca^{2+} concentrations and consider testing other lipid compositions to assess the consistency of their observations. Alternatively, they could exclude these results from the manuscript given that they are unnecessary to support the central message of the manuscript, which is that Ssp4 assembles ion-selective pores.

This study is not intended to be a biophysical characterisation of the pore. The Ca^{2+} effect that we observe is consistent, reproducible, interesting and important. There is no justification to remove these data from the manuscript.

Finally, the study seems limited to a single lipid composition (PE), as indicated in the methods. Exploring additional lipid mixtures would provide valuable insights into how the lipid environment influences pore formation and function.

Yes, we agree this would be an interesting study for the future, but it is not within the scope of the current study.

This study is not intended to be a biophysical characterisation of the pore. We are simply demonstrating that Ssp4 forms regulated ion-conducting pores, thereby assigning function to a novel family of T6SS-delivered effector proteins.

1. Queralt-Martín M, López ML, Aguilera-Arzo M, Aguilera VM, Alcaraz A: Scaling Behavior of Ionic Transport in Membrane Nanochannels. *Nano Lett* 2018, 18:6604-6610.
2. Andersen OS, Koeppe II RE, Roux B: Gramicidin Channels: Versatile Tools. In *Biological Membrane Ion Channels*. Springer New York; [date unknown]:33-80.
3. Hille B: *Ionic channels of Excitable Membranes* 2nd edition. *J Electrochem Soc* 1987, 134.
4. Aguilera VM, Queralt-Martín M, Aguilera-Arzo M, Alcaraz A: Insights on the permeability of wide protein channels: measurement and interpretation of ion selectivity. *Integr Biol* 2011, 3:159-172.
5. Melcrová A, Pokorna S, Pullanchery S, Kohagen M, Jurkiewicz P, Hof M, Jungwirth P, Cremer PS, Cwiklik L: The complex nature of calcium cation interactions with phospholipid bilayers. *Sci Rep* 2016, 6:38035.
6. Marra J, Israelachvili J: Direct measurements of forces between phosphatidylcholine and phosphatidylethanolamine bilayers in aqueous electrolyte solutions. *Biochemistry* 1985, 24:4608-4618.

Review Structural modelling and molecular dynamics simulation:

As pointed out by the authors, "Upon delivery to their target cellular compartment, α -PFT monomers typically undergo a conformational change resulting in exposure of hydrophobic or amphipathic helices, followed by oligomerisation and concomitant membrane insertion to form the mature pore³⁴. Such toxins are thus well suited to delivery by the T6SS, where two forms of the protein are likely to exist: a pre-secretion form which is soluble and compact to allow loading into the Hcp tube, and an active form with exposed hydrophobic helices primed for membrane insertion."

Current limitations of AlphaFold (AF) for predicting the structure of pore-forming toxins include:

- AF struggles with accurately predicting the membrane insertion steps of pore-forming toxins, as these involve dynamic conformational changes that are not easily captured by static structural predictions.
- Furthermore, predicting oligomeric structures, such as tetramers or larger assemblies, is particularly difficult. The dynamic process of oligomerization, crucial for pore formation, is challenging to model accurately with current AF capabilities.
- The influence of the lipid environment on toxin structure and function is challenging to integrate into AF predictions. Factors such as lipid composition and membrane dynamics are critical for pore formation but are not accounted for in AF's modeling process.
- The accuracy of AF predictions relies heavily on the availability of diverse and extensive Multiple Sequence Alignments (MSAs). For many pore-forming toxins, especially those without well-characterized homologs, this can be a significant limitation.

We thank the Reviewer for their extensive feedback (both above and below) on this section of the manuscript. We agree with the Reviewer that there are many limitations to using AlphaFold to predict the structure of pore-forming and other membrane proteins. This is why in the original manuscript, and even more so in the revised version, we have aimed to assess the accuracy of our AlphaFold models, and additionally gain further insight into the potential mechanism of ion conductance, by using molecular dynamics simulation analysis and comparison with our experimental electrophysiology data. Importantly, however, this Reviewer and Reviewer 3 were correct to question the strength of our evidence for Ssp4 being a tetramer. Their comments led to us re-evaluating our initial assumption that this was the case, performing further analyses, and providing a better-supported conclusion.

In our updated analysis, we took all oligomeric forms for which AlphaFold generated predictions with plausible structures for a membrane complex, namely monomer to octamer. Rather than select one of these as the preferred form based on simpler tools and some assumptions, and then analyse that form by molecular dynamics simulations (MD), as we did previously, this time we took all eight forms and tested them for stable and non-disruptive membrane insertion using MD, a more robust method. This narrowed down the possible membrane-integrated Ssp4 structures to four, monomer to tetramer. Then we tested each of these (rather than just the tetramer as previously) for their ability to conduct ions using MD, yielding the interesting outcome that all four forms are predicted to contain hydrated water channels/pores which will allow them to conduct ions. This makes sense, because these hydrated pores are within-monomer and form similarly independent of oligomeric

state, although only one monomer appears to be actively conducting ions at a time in multimeric forms.

This more thorough analysis means that we cannot definitively predict in which of these forms Ssp4 exists; indeed more than one form may exist simultaneously or in response to differing membrane conditions. The simplest, and perhaps most likely, scenario is that Ssp4 may simply work as a monomer. We have made this clear in the revised manuscript. Importantly, the basic mechanism and pathway of ion conductance through a single monomer suggested by the MD simulations is strongly supported by our experimental data, where we observed a very similar conductance value in the lab as that generated in silico. This would not happen by chance. We also note that this is unchanged from our original observations – conductance was also occurring through a single subunit there, we had just incorrectly assumed that it was always within the context of a tetramer.

We believe that this part of the manuscript is now stronger and more accurate, is explained more clearly, and provides some interesting initial insight into the function of Ssp4 that will lead future studies of this protein and others. Further specific points not covered above are answered below.

How do the authors interpret AF results? What conformation is AF predicting for the Ssp4 monomer, the soluble, or the membrane form with exposed hydrophobic helices?

We have now explained in the Discussion that we believe that AlphaFold is predicting the membrane form and that Hcp is likely to act as a chaperone to stabilise a different, soluble conformation prior to secretion (we have shown that Hcp stabilises Ssp4 previously).

Do they observe significant conformational differences between the AF-predicted monomeric and tetrameric structures?

No. This is now included in the manuscript (including RMSD values in Appendix Table S1).

What are the predicted aligned error (PAE) scores for the tetramer? What are the scores for the regions involved in protein-protein interactions?

Given the altered focus and conclusions of the revised manuscript, we have included plots of PAE values for all the multimers in Appendix Table S1.

Do they observe significant conformational changes in Ssp4 following the 1-microsecond MD simulations of the AF tetramer in a POPC model membrane?

We do not observe any major conformational changes.

In general, bacterial membranes are dominated by phosphatidylethanolamine (PE), phosphatidylglycerol (PG), and cardiolipin (CL). So, why did they choose to carry out MD simulations on a POPC model membrane?

We apologise for writing POPC in error, it was actually a POPE model membrane; this has now been corrected. (Charged PG complicates the simulation system, while CL is very ill-behaved in simulations and usually not used when modelling bacterial membranes; additionally we used uncharged lipids in the experimental work).

Have they attempted to validate experimentally that Ssp4 assembles tetrameric pores, for example, by analytic ultracentrifugation or native Mass Spectrometry of Ssp4 pores embedded in detergent micelles?

First, we note that we are no longer claiming that Ssp4 necessarily forms tetrameric pores, so this is no longer a definitive conclusion that requires validation.

We agree it would be nice to experimentally determine the oligomeric form, or forms, adopted by Ssp4 in a membrane environment, but this is not a trivial undertaking and we have not done so to date. Ssp4 oligomers would need to be extracted from membranes to be sure that the correct form was being analysed (addition of soluble protein to detergent micelles will likely not be able to reproduce the physiologically-relevant situation). However when Ssp4 does integrate into bacterial membranes, for example when expressed in *E. coli* using an N-terminal signal peptide, it is highly toxic to the cells, prohibiting overproduction and purification of the protein in this context. We appreciate that there may be approaches such as native mass spectrometry or cryoEM analysis of Ssp4 embedded in membrane-like environments that could determine oligomeric state, but the level of time and expertise required is outside of the scope of this study.

In the legend of Fig. 5c, the authors present an overlay of K⁺ ion positions from a 1 μ s simulation, suggesting a preferred ion permeation pathway across the pore formed by Ssp4. However, in the discussion section, the authors state: "Our modeling suggests that Ssp4 forms tetrameric pores with water-filled, ion-conducting channels..." In their MD simulations, do they observe ionic transport through several channels within the Ssp4 tetramer? What are the diameters of these channels? Do these diameters correlate with those estimated by their electrophysiology measurements and those of gramicidin A?

We observe ion conduction only through one pore at a time with a conductance value and ion selectivity that agree with our electrophysiology measurements. Gramicidin is a single-file water channel, that is, its diameter is just wide enough to accommodate single water molecules along the pore axis, whereas the Ssp4 pore displays a diameter of about 5Å and is occupied by several water molecules at each position on the pore axis (see Fig. 5).

What residues line the lumen of the Ssp4 tetrameric channel? To validate their model, they could consider mutating some of these residues and measuring the effect on ion conductance.

As is now explained more clearly in the manuscript, and elsewhere in this response, there is not an ion-conducting channel with a lumen formed in the centre of the Ssp4 tetramer (or any other oligomer). Ssp4 is not a classical ion channel in this respect, as our MD simulations have revealed. However, all individual subunits of the oligomers and the monomer each form a water-filled pore, which is lined by an abundance of polar threonine, serine, and asparagine residues and contains aspartic acid side chains as well (see, e.g., new Appendix Figure 5).

Considering the limitations of AF for predicting the structures of PFTs and the lack of experimental validation of the AF model, their computational conclusions need to be reevaluated. For example, I would strongly disagree with the following statement: "Here, we present the first high confidence structural prediction for such a pore". What makes them think they obtained a high-confidence structural prediction of Ssp4 pore?

We have removed the phrase 'high-confidence'. As above, our original conclusion about the oligomeric state of the pore has been revised and is no longer so definitive. We now present the more nuanced conclusion that the Ssp4 pore could exist as one or more of several possible oligomeric forms, perhaps most likely the monomeric form, but the mechanism of ion conductance is not dependent on oligomeric state, since the ions do not flow through a central channel formed by subunit oligomerisation. We believe that we can have confidence in the predicted structure of the Ssp4 monomer due to the strong agreement between the ion conductance values generated by the molecular dynamics simulations and the experimental electrophysiology data.

Referee #2:

In this manuscript, the authors show that T6SS toxins Ssp4 and Ssp6 are pore-forming toxins with wide range and narrow range of targets, respectively. Ssp4 acts from the periplasm, kills target cells without their lysis and has a transmembrane topology. The authors show that Ssp4 depolarizes the membrane by forming pores (MD simulation and measurements) and that this also leads to ROS production. The authors further performed a Tn-seq experiment to identify genes required for resistance and sensitivity to Ssp4 in *P. fluorescens*. This and subsequent mutagenesis identified that deletion of C-term of *mucA* provides protection against Ssp4 but also other toxins. Deletion of *moaC* provided a relatively modest protection as well. This is a manuscript with high-quality data and figures as well as clear explanations. The experiments are logical, with sufficient replicates, and well-controlled as well as properly analyzed and interpreted.

We thank the Reviewer for their positive comments and suggestions.

Comments:

It would be interesting to delete more genes in the same pathway as *moaC* to test if the observed phenotype is dependent on *moaC* in particular or rather on the whole pathway.

We agree that this would be interesting but feel it should be explored fully in a follow-up study rather than the current work.

For the *mucA* truncation mutant, it would make sense to test if an additional deletion of a gene in the EPS synthesis pathway would restore the toxin sensitive phenotype.

We thank the reviewer for this excellent suggestion. We have now included new data in the manuscript where we introduced a deletion of *algD* (*AlgD* is essential for alginate EPS biosynthesis) into the *mucA*_{Δ76-195} mutant in order to determine whether alginate production is responsible for the T6SS resistance phenotype. This gave the unexpected result that alginate production is not required for T6SS resistance (since the *mucA*_{Δ76-195} Δ*algD* mutant was as resistant as the *mucA*_{Δ76-195} mutant), meaning that another trait regulated by *MucA* is responsible. This is an exciting result because this represents a new mechanism of immunity-independent T6SS resistance, distinct from those reported previously. Defining this new mechanism is outside the scope of the current study but we have noted in the discussion some interesting observations from metabolic and transcriptomic analysis of a *P. fluorescens mucA* mutant which are likely to be relevant. We have revised the Discussion section to take account of this new data and insight.

This is not essential, but could you try to identify more resistant mutants by expressing Ssp4 in the periplasm of target cells rather than by mixing them with the T6SS+ attacker?

Unfortunately, our experience suggests that we would select regulatory and point mutants that no longer express sp-Sp4 rather than those that are resistant to the action of Ssp4.

While you clearly show that Ssp4 targets bacterial membrane, is there any evidence that it fails to target eukaryotic membranes?

We have not tested this directly, but we have shown, for example, that Ssp4 does not contribute to the antifungal activity of the Db10 T6SS (DOI: [10.1038/s41564-018-0191-x](https://doi.org/10.1038/s41564-018-0191-x)). We believe that it may be restricted to bacterial targets due to a requirement to enter the membrane from the periplasmic compartment and/or its requirement for DsbA-mediated disulphide bond incorporation (DOI: [10.1016/j.celrep.2017.12.075](https://doi.org/10.1016/j.celrep.2017.12.075)).

Referee #3:

In this manuscript, Reglinski et al. perform a comprehensive analysis of the *Serratia* T6SS effector Ssp4. They use various genetic, biochemical, and biophysical techniques to investigate its toxicity, target range, mechanism of action, and distribution. Furthermore, they use a Tn-seq approach to identify a mechanism that enables target protection against Ssp4 intoxication. Overall, this is an interesting, well-written paper. The flow and rationale are clear, and most conclusions are well supported by the results. I have a few comments for the authors to consider.

We thank the Reviewer for their positive comments and suggestions.

Major comments:

1. The authors suggest that Ssp4 and Ssp6 are investigated "in isolation" when expressed in a D9 strain (lines 117-119). However, there appears to be a significant difference in intoxicating either *E. coli* or *P. fluorescens* targets when comparing the *tssE* mutant and D9 attacker strains. This suggests that additional effectors are delivered by this T6SS. Therefore, the conclusions could be toned down and discussed in the light of a possible synergistic effect of Ssp4 or Ssp6 with other effectors, since it is possible that they are not truly examined individually in the D9 strain.

The Reviewer is correct that we cannot exclude the possibility that there is synergy between Ssp4 or Ssp6 and an additional effector(s) when intoxicating non-Db10 target cells with $\Delta 9 + \text{Ssp4}$ or $\Delta 9 + \text{Ssp6}$. However any effects that are observed to be different between $\Delta 9$ and $\Delta 9 + \text{Ssp4}$, or between $\Delta 9$ and $\Delta 9 + \text{Ssp6}$, must be dependent on Ssp4 or Ssp6, respectively. Therefore we can conclude, for example, that Ssp6 is able to act against some target species but not others, even if the magnitude of the effect observed could theoretically be influenced by synergy with an unknown effector.

In order to address this point, we have changed the wording in the place mentioned to remove the phrase 'in isolation' and we have added a short section to note the potential existence of additional effectors which could synergise with Ssp4 or Ssp6 when delivered into heterologous target species by $\Delta 9 + \text{Ssp4}$ or $\Delta 9 + \text{Ssp6}$.

Apart from Figure 2, the only other part of the manuscript where we used $\Delta 9$, $\Delta 9 + Ssp4$ and $\Delta 9 + Ssp6$ in competition against a non-Db10 target species was during our attempt to identify resistant mutants of *P. fluorescens* by Tn-seq and subsequent assessment of the resistance phenotype of clean mutants in this species. In this case, we had already highlighted the potential contribution of the ‘delivery of an as-yet-unidentified effector’ in the corresponding part of Discussion section.

2. lines 123-125: this is an over-reaching conclusion to base on only 2 target strains. On the same topic - Lines 307-308 and 310-312: To draw this conclusion and make these claims, the author should test more than 2 target strains.

In the first version of the manuscript and our previous study, we tested the ability of Ssp4 and Ssp6 to intoxicate three target species, *Serratia marcescens*, *E. coli* and *P. fluorescens*. We agree with the Reviewer that our conclusion, namely that Ssp6 may only target species more closely-related to *S. marcescens* whereas Ssp4 acts broadly against a wide range of species, would be more robust if further target species are tested. We have therefore included new data using *Enterobacter cloacae* (an additional member of the Enterobacterales) and *Burkholderia thailandensis* (non-Enterobacterales, more distantly related than *Pseudomonas*) as target strains. This shows that Ssp6 does act against *Ent. cloacae* but does not act against *B. thailandensis*, whilst Ssp4, again, acts against both (Figure 2, new panels d and e). Therefore we have now tested five target species in total, with the new data supporting our original conclusion.

3. Fig. 3b: ~7% of the cells being affected seems like a very low percentage, especially when the effector is expected to be over-expressed in all these cells. Can the authors comment on that? Also, did the authors attempt to use a TAT signal to send the effectors to the periplasm? Is it possible that Ssp6 is not toxic when ectopically expressed in *P. fluorescens* because it is not properly transported and folded in the periplasm when it passes unfolded through the Sec system?

We would like to highlight that this is not plasmid-based overexpression but rather single-copy chromosomal expression, therefore it is not directly comparable with studies using high level induction of genes on multicopy plasmids. It is also possible that the model signal peptide we used for both effectors, from *E. coli* OmpA, was not highly efficient in *P. fluorescens*.

We can reassure the Reviewer that Ssp6 is not incompatible with passing unfolded through the Sec system, since we observed toxicity, bacteriostasis, membrane depolarisation and loss of OM integrity on expression of the same sp-Ssp6 fusion protein, prevented by co-expression of Sip6, in our previous study (DOI: 10.1038/s41467-019-13439-0). We also note that the lack of overt toxicity on ectopic expression of sp-Ssp6 in *P. fluorescens* is consistent with what is seen upon T6SS delivery of the native protein into the same species.

4. The section on structure and fold analyses of Ssp4 (starting at line 174) is rather weak and speculative in my view. The MD analysis is based on a biased assumption that Ssp4 functions as a tetramer, which itself is based on a predicted structure that is not similar to any known and proven structure. I think that without further experimental data supporting a tetrameric complex, this section is too speculative. If the authors choose to keep it as part of the work, I suggest moving it to the supplementary materials and discussing the biases and caveats more thoroughly.

Please see the response to Reviewer 1. In short, we took on board both Reviewers’ comments questioning the evidence for a tetrameric pore and went back and performed further analyses and

simulations. This has allowed us to present the more nuanced conclusion that the Ssp4 pore could exist as one or more of several possible oligomeric forms, but the mechanism of ion conductance is not dependent on oligomeric state, since the ions do not flow through a central channel formed by subunit oligomerisation. We believe that we can have confidence in the predicted structure of the Ssp4 monomer due to the strong agreement between the ion conductance values generated by the molecular dynamics simulations and the experimental electrophysiology data.

5. MucA was identified in the screen as protecting the target cells from Ssp6. It is unclear:

A) why would this happen if Ssp6 isn't toxic to this target to begin with.

B) Why was this "protection" not observed in the competition assay in Suppl. Fig. 8?

C) Did the authors consider a polar effect of the Tn mutation?

We believe that points (A) and (B) are likely explained by the high sensitivity of the TnSeq approach coupled with differences in the experimental set-up between this experiment and the standard competition assay. The *mucA* mutant may have a detectable fitness advantage under conditions of Ssp6 delivery in the Tn-seq experiment and not in the standard competition setting because in the former, each *P. fluorescens* genotype/mutant has to compete against all the other *P. fluorescens* mutants, but in the latter it does not. If a sub-lethal stress is imparted by delivery of Ssp6, or another indirect consequence on T6SS activity in the $\Delta 9$ +Ssp6 strain compared with $\Delta 9$, the *mucA*-disrupted mutants outcompete the *mucA*-intact mutants. This slight competitive advantage is not detectable in a standard competition assay.

In the original manuscript we already had a paragraph in the Discussion noting that the selection of *mucA* mutants in the Tn-seq comparison of $\Delta 9$ + *ssp6* vs $\Delta 9$ was unexpected and proposing two explanations for why delivery of Ssp6 might impose a small selective pressure/fitness decrease detectable by the Tn-seq assay ('It is possible that interaction of Ssp6 with the membrane in *P. fluorescens* leads to rare or immature pores conferring a small fitness defect detectable by this assay. Alternatively, reintroduction of Ssp6 into the $\Delta 9$ mutant might increase T6SS firing rate via a checkpoint for effector loading, leading to increased T6SS damage to the target cells, either from penetration or delivery of an as-yet-unidentified effector.'). We have slightly revised this paragraph to ensure clarity and have now specifically noted that a small fitness advantage caused by truncation of MucA in the presence of $\Delta 9$ + *ssp6* may only be apparent when the *mucA* mutant is directly competing against other genotypes of *P. fluorescens* as is the case during the Tn-seq selection.

For point (C), even if the Tn mutant was polar and affecting other genes, that would not provide an answer as why it would be selected for under conditions of *ssp6* intoxication vs control if Ssp6 is not overtly toxic. We did consider polarity, which is why we made the clean in-frame deletion for subsequent experiments.

Minor comments:

1. Line 131: it is unclear whether these are 15-20% of the target cell population or the entire population (attackers+targets).

We have changed 'total population' to 'total population (attacker and target cells)'.

2. Fig. 3c: Ssp6 appears to associate with the membrane without being toxic. Will every protein that is sent to the periplasm using a signal-peptide be associated with the membrane fraction in this assay? What I mean is that perhaps membrane association is an artifact in this case?

We thank the Reviewer for highlighting this possibility. We repeated the assay using a new strain expressing a catalytically-inactive variant of a peptidoglycan amidase effector, which is predicted to be a soluble periplasmic-acting effector, fused to a signal peptide in the same manner as Ssp4 and Ssp6. This effector could also be detected in our 'membrane' fraction. This result, coupled with our lack of a full set of compartment controls in this organism, means that we cannot be fully confident that the appearance of sp-Ssp4 or sp-Ssp6 in our 'membrane' fraction accurately reflects genuine membrane association. Therefore we have removed this panel and associated discussion from the manuscript. Our electrophysiology experiments provide separate, direct evidence that Ssp4 interacts directly with membranes.

3. Can the authors please explain why the ectopic expression of Ssp4 and evaluation of its effect on membrane depolarization were done in *P. fluorescens*, whereas the ROS assays are performed by expressing Ssp4 in *E. coli* cells?

The assay to evaluate membrane depolarisation assay upon ectopic expression was performed in *P. fluorescens* in order to compare with the unexpected observation that Ssp6 did not show detectable anti-bacterial activity in *P. fluorescens* under conditions of T6SS delivery (in the co-culture assay). The ROS assay was performed in *E. coli* so that the impact of Ssp4 and Ssp6 could be compared in a context where both showed toxicity/antibacterial activity.

4. Fig. 6: The effect of gentamicin, while possibly passing the specific statistical test at the 3-h time point, is minor. I am not sure how it adds to the story. Furthermore, the effect of Ssp6 in Fig. 6c appears comparable to that of gentamicin at this time point. Did the authors test for statistical significance between the 0 and 3-h time points in the Ssp6 samples?

Gentamicin was the positive control and reference point for this assay when we did not know if either effector would cause ROS production. The data show that ROS production in response to intoxication by Ssp4 is more significant than the relatively small but previously-reported effect of gentamicin.

Figure 6c is a different experiment from Figure 6b, therefore whilst the data are consistent between the two experiments, the values cannot be directly compared.

The difference between the 0 h and 3 h time points is not statistically significant for either sp-Ssp6 or sp-Ssp6 + Sip6. We agree that there appears to be a very small amount of ROS production in sp-Ssp6 at 3 h compared with control samples, but since there was no significant difference between sp-Ssp6 and sp-Ssp6 + Sip6 we cannot conclude that there is any specific effect due to Ssp6.

5. Lines 217-227: This section leads to more questions than answers. I don't think it contributes much to the manuscript, and the authors might consider removing it.

We have chosen to retain this section because although the data are somewhat 'negative', it provides two important insights to the field: (1) a caution that the Cpx system is not necessarily important or required for ROS-dependent toxicity, despite suggestions in the literature, and (2) further evidence that the Cpx system, despite being an envelope stress response triggered by membrane disruption

which might therefore be predicted to be involved in resistance against T6SS attacks (DOI: 10.1016/j.celrep.2020.108259), at least in this context, does not contribute to T6SS resistance. However we accept the Reviewer's point that this is not a central message of the paper and so have moved panel Fig. 6D to Supplementary Figure 6, now Fig. EV1.

6. Line 239: is the term "Ssp6 intoxication" appropriate in light of the observation that Ssp6 is not toxic to *P. fluorescentis*?

This term was initially used for ease of reading in comparison with Ssp4 intoxication. However the Reviewer is correct that it is not really appropriate, so we have revised the sentence to remove it.

7. Line 244: Can the authors please clarify why they chose these 2 genes and not others?

They were chosen because they had the largest fold-change and $-\log_{10}$ p-val values in the $\Delta 9 + \text{Ssp4}$ vs control ($\Delta 9$) comparison (ie mutants in these genes were the most substantially enriched or depleted in this comparison, in both cases depleted). A comment to this effect has been added to the sentence.

8. A list of the identified Ssp4 homologs should be provided as supplementary material. It would also be informative to determine whether the homology includes a conserved motif or structural features, such as the same number of TM helices. Please also specify in the Methods section what was the coverage cutoff when deciding what is a likely Ssp4 homolog, and whether it always included the TM region of the proteins?

A list of the identified Ssp4 homologues is now provided as Dataset EV2. This details the UniProt accession, the start and end of the hit defined by hmmsearch, and the hmmsearch score. The Ssp4 model covered almost the entire length of Ssp4 and included all four predicted TM helices. We did not include a coverage cut off, in order to not exclude protein fragments and maximise taxonomic coverage. However almost all (95%) of the identified homologues showed homology across the majority of the Ssp4 sequence (>200 amino acids), including all four TM helices. We have now included an amino acid sequence alignment of all the identified homologues showing the position of the predicted transmembrane helices (showing that these regions are well conserved) as a new Appendix Figure S6 to illustrate these points and have included the relevant information in Methods.

9. Are the predicted Ssp4 homologs immunity proteins similar, or are there different families?

There appears to be some variation, but we have not performed the analysis required to define different families.

10. Lines 305-307: Similar observations were made previously with other T6SS effectors. Perhaps it is appropriate to discuss these as well?

We have now included a short section and references citing the relevant papers.

11. It would be nice to see that MucA complementation re-sensitizes the target cell to Ssp4 intoxication.

We had not done this previously because it is not as straightforward as complementing a complete gene deletion with an intact copy of the gene. Our MucA mutant represents a deletion of the C-terminal domain of the protein (as a full gene deletion would be lethal). Therefore by supplying an

intact (ie full-length) copy of the gene, we are generating a situation where there is the incorrect stoichiometry of N-terminal to C-terminal domains (2:1 instead of 1:1) and not restoring the wild type situation. Expressing only the C-terminal domain as a separate protein would also not recapitulate the wild type protein. Given the broad and finely-tuned regulatory impact of bacterial sigma factors (including AlgU, regulated by MucA) and the importance of MucA in *Pseudomonas* physiology, we considered it unlikely that such a complementation would work to restore a wild type phenotype. Indeed, when we did perform this experiment, on the request of the Reviewer, we did not see restoration of a wild type phenotype:

Figure R2. *Ssp4*-dependent anti-bacterial activity against *P. fluorescens* with wild type MucA, truncated MucA (*mucA*_{Δ76-195}), or truncated MucA with full length MucA expressed in trans.

Importantly, we already have strong evidence that the phenotype of our MucA truncation mutant is not as a result of a secondary mutation elsewhere in the genome, since our Tn-seq data (where multiple independent *mucA* C-terminal insertions were selected) combined with our clean in-frame deletion of the *mucA* C-terminus represent multiple independent observations of the T6SS-resistance phenotype.

We note that we cannot exclude the possibility that, even in our clean in-frame deletion of the MucA C-terminus, the expression of MucB, encoded immediately downstream of *mucA*, has also been affected in our *mucA* mutants due to translational coupling (although the deletion was designed to avoid any impact on MucB translation). This might also explain why the complementation failed. However, it does not have any impact on our conclusions, since MucB is the partner protein of MucA, together forming the anti-sigma factor for AlgU. Indeed the role of MucB is to bind to the C-terminal domain of MucA that is lost or deleted in our MucA mutants, so any loss of MucB is not expected to have additional impacts on top of these mutants.

12. Lines 404-406: Do the *mucA* mutants form mucoid colonies?

The *mucA* mutant does not form visibly mucoid colonies in the conditions used for the co-culture assays, which is consistent with our new observation that the T6SS resistance associated with this mutant is independent of alginate production.

Dear Sarah,

Thank you for submitting a revised version of your manuscript. We have now received input from two of the original reviewers. While reviewer #3 is satisfied with the revisions, reviewer #1 indicates a number of remaining concerns that should be addressed with textual clarifications and toning down of the statements. There are also a few editorial points that need to be addressed before I can extend official acceptance of the manuscript:

1. Please submit up to five keywords.
2. Please make sure that the order of the sections in the manuscript is as follows: abstract, introduction, results, discussion, materials & methods, data availability section, acknowledgments, disclosure and competing interests statement, references, main figure legends, tables, expanded view figure legends.
3. CRediT has replaced the traditional author contributions section because it offers a systematic, machine-readable author contributions format that allows for more effective research assessment. Please remove the Authors Contributions from the manuscript and use the free text boxes beneath each contributing author's name in our online submission system to add specific details on the author's contribution. More information is available in our guide to authors.
4. Individual figure panels for Fig. EV1 and EV2 are not mentioned in the manuscript text. Please add the corresponding callouts.
5. Figure 6D is mentioned in the manuscript text, but is absent in the figure file - please check.
6. Please add the EV dataset legends to the corresponding Excel files in a separate tab/worksheet.
7. Our data editors have flagged the following issues in figure legends that need correcting:
 - Please provide the exact p values in the legends of figures 2A-E; 3A, B; 6B, C; 7C, D, E, F; EV1 A, EV3 C.
 - Please indicate the statistical test used for data analysis in the legend of figure EV2 A.
8. Papers published in The EMBO Journal are accompanied online by a 'Synopsis' to enhance discoverability of the manuscript. It consists of A) a short (1-2 sentences) summary of the findings and their significance, B) 3-4 bullet points highlighting key results and C) a synopsis image that is 550x300-600 pixels large (width x height, jpeg or png format). You can either show a model or key data in the synopsis image. Please note that the image size is rather small and that text needs to be readable at the final size.

With best wishes,

Ieva

We realize that it is difficult to revise to a specific deadline. In the interest of protecting the conceptual advance provided by the work, we recommend a revision within 3 months (17th Nov 2025). Please discuss the revision progress ahead of this time with the editor if you require more time to complete the revisions.

Referee #1:

The authors should be commended for their comprehensive and scientifically rigorous response to the major critiques from the

initial review. The manuscript has been significantly improved as a result. The most substantial revision—a comprehensive re-evaluation of the Ssp4 toxin's oligomeric state and mechanism—represents a worthy scientific pivot. The abandonment of the weakly supported tetramer-only model in favour of a more nuanced, data-driven hypothesis involving a novel "within-monomer" conducting pore has substantially strengthened the manuscript and increased its potential impact. The authors have also appropriately addressed several other critiques, including refining the terminology for ion selectivity and employing the correct theoretical framework for analysing divalent cation permeability.

However, despite these significant improvements, several key biophysical arguments presented in the rebuttal and revised manuscript require further attention to ensure the conclusions are fully supported by sound physical reasoning. The central validation for the new structural model—a claimed "strong agreement" between simulated and experimental ion conductance—is weakened by a failure to account for the different ionic concentrations used. More critically, a key comparison used to rationalise the observed conductance is based on a biophysically inconsistent premise, and the description of the calcium permeability experiments is inaccurate.

The work represents a valuable contribution to the field of bacterial competition and T6SS effectors. Acceptance is recommended following a final round of minor, but essential, revisions that address the identified issues of biophysical interpretation.

Minor points for revision

1. The claim of "Strong agreement" between MD and experimental conductance is overstated and requires qualification. The authors' primary validation for their new structural model is the supposed "strong agreement" between the ion conductance calculated from MD simulations and the value measured in their electrophysiology experiments (Manuscript p. 13, lines 392-394). They present the following values:

- Experimental value: 18.4 {plus minus} 0.64 pS in symmetrical 510 mM KCl (Manuscript p. 7, lines 169-170).
- MD Simulation value (Tetramer): 15 {plus minus} 6 pS in 150 mM KCl (Manuscript p. 8, lines 219-221).
- MD Simulation value (monomer): 4 {plus minus} 3 pS in 150 mM KCl (Manuscript p. 9, lines 238-239).

A direct comparison of these numbers is misleading. Single-channel conductance is highly dependent on the concentration of the permeant ion; for a simple pore, conductance generally increases with ion concentration until saturation effects become dominant. Therefore, the conductance measured in 510 mM KCl is expected to be substantially higher than that in 150 mM KCl. That is the case of MD simulation value for the monomer, but in the case of the tetramer the calculated and experimental values are similar, suggesting that the simulation for the tetramer may be overestimating the intrinsic conductance of the pore.

Recommendation: The claim of "strong agreement" should be revised to accurately reflect the significant difference in ionic strength. To make a truly compelling case, the authors should perform new MD simulations under ionic conditions that match the *in vitro* experiments (510 mM KCl). At a minimum, this significant caveat must be explicitly acknowledged and discussed in the manuscript to avoid overstating the validation of their computational model.

2. Inaccurate description of calcium permeability experiments.

The authors state that when CaCl_2 is in the trans chamber, "rectification was apparent at voltages > 50 mV" (Manuscript p. 7, lines 183-185). However, a close analysis of the provided raw data reveals this description to be inaccurate (Raw Data; Manuscript Fig. 4D). Rectification implies a voltage-dependent change in conductance where current continues to increase with voltage. The data, however, show that the current plateaus and ceases to increase at positive potentials. For instance, in one replicate, the current remains completely flat at 0.763 pA for all voltages from +50 mV to +80 mV. This phenomenon is more accurately described as current saturation.

This distinction is mechanistically important. Saturation suggests a rate-limiting step in permeation.

Recommendation: The authors should revise their description of the data in Figure 4C,D from "rectification" to the more accurate and mechanistically informative term "saturation."

3. Potential confounding variables in calcium experiments remain unaddressed.

While the authors have correctly recalculated the $\text{Ca}^{2+}/\text{K}^{+}$ permeability ratio, they have dismissed the valid concern about the well-documented effects of Ca^{2+} ions on the physical properties of the phosphatidylethanolamine (PE) membranes used in their experiments. Their justification that this is "not intended to be a biophysical characterisation of the pore" and is "outside the scope of the current study" is insufficient when they proceed to interpret their data as evidence for a direct " Ca^{2+} -dependent conformational change" within the Ssp4 protein itself (Manuscript p. 7, lines 189-190). Even with the more precise interpretation of current saturation, which points strongly to a protein-mediated effect, the authors cannot attribute the overall change in conductance solely to a protein conformational change without ruling out membrane-level effects.

Recommendation: The interpretation of a direct conformational change in the protein should be significantly toned down, and the potential for membrane-level effects should be explicitly discussed as a confounding variable.

Conclusion

The authors have been exceptionally responsive to the initial reviews, and their revised manuscript presents a novel and exciting model for the function of the Ssp4 pore-forming toxin. The work is of high quality and potential impact. My recommendation is for acceptance following minor revisions that address the essential points of biophysical interpretation detailed above. Ensuring the manuscript's claims are supported by sound physical reasoning will solidify its conclusions and maximize its impact on the field.

Referee #3:

The authors addressed all of my concerns. I congratulate them on this elegant work.

Response to Reviewer Comments (EMBOJ-2025-120361R)

Referee #1:

The authors should be commended for their comprehensive and scientifically rigorous response to the major critiques from the initial review. The manuscript has been significantly improved as a result. The most substantial revision—a comprehensive re-evaluation of the Ssp4 toxin's oligomeric state and mechanism—represents a worthy scientific pivot. The abandonment of the weakly supported tetramer-only model in favour of a more nuanced, data-driven hypothesis involving a novel "within-monomer" conducting pore has substantially strengthened the manuscript and increased its potential impact. The authors have also appropriately addressed several other critiques, including refining the terminology for ion selectivity and employing the correct theoretical framework for analysing divalent cation permeability.

However, despite these significant improvements, several key biophysical arguments presented in the rebuttal and revised manuscript require further attention to ensure the conclusions are fully supported by sound physical reasoning. The central validation for the new structural model—a claimed "strong agreement" between simulated and experimental ion conductance—is weakened by a failure to account for the different ionic concentrations used. More critically, a key comparison used to rationalise the observed conductance is based on a biophysically inconsistent premise, and the description of the calcium permeability experiments is inaccurate.

The work represents a valuable contribution to the field of bacterial competition and T6SS effectors. Acceptance is recommended following a final round of minor, but essential, revisions that address the identified issues of biophysical interpretation.

We thank the Reviewer for their positive comments and appreciation of the efforts we made in revising the manuscript.

Minor points for revision

1. The claim of "Strong agreement" between MD and experimental conductance is overstated and requires qualification.

The authors' primary validation for their new structural model is the supposed "strong agreement" between the ion conductance calculated from MD simulations and the value measured in their electrophysiology experiments (Manuscript p. 13, lines 392-394). They present the following values:

- Experimental value: 18.4 {plus minus} 0.64 pS in symmetrical 510 mM KCl (Manuscript p. 7, lines 169-170).
- MD Simulation value (Tetramer): 15 {plus minus} 6 pS in 150 mM KCl (Manuscript p. 8, lines 219-221).
- MD Simulation value (monomer): 4 {plus minus} 3 pS in 150 mM KCl (Manuscript p. 9, lines 238-239).

A direct comparison of these numbers is misleading. Single-channel conductance is highly dependent on the concentration of the permeant ion; for a simple pore, conductance generally

increases with ion concentration until saturation effects become dominant. Therefore, the conductance measured in 510 mM KCl is expected to be substantially higher than that in 150 mM KCl. That is the case of MD simulation value for the monomer, but in the case of the tetramer the calculated and experimental values are similar, suggesting that the simulation for the tetramer may be overestimating the intrinsic conductance of the pore.

Recommendation: The claim of "strong agreement" should be revised to accurately reflect the significant difference in ionic strength. To make a truly compelling case, the authors should perform new MD simulations under ionic conditions that match the in vitro experiments (510 mM KCl). At a minimum, this significant caveat must be explicitly acknowledged and discussed in the manuscript to avoid overstating the validation of their computational model.

We thank the Reviewer for highlighting that the manuscript would benefit from explicitly mentioning this point, whilst also noting that our data indicate that the impact of the difference in ion concentrations used is unlikely to be as dramatic as the Reviewer envisages.

We also determined experimental conductance values for Ssp4 under gradient (asymmetric) conditions (510 mM:210 mM KCl). These values are now included in the revised manuscript (lines 169-170). Since the experimental conductance is comparable between 510:210 mM and 510:510 mM (19.4 +/- 0.93 pS and 18.4 +/- 0.64 pS, respectively), we conclude that we are close to the maximal single channel conductance, i.e. saturation, at 210 mM KCl (DOI: 10.1085/jgp.76.4.425), with the simulation conditions (150 mM) being at only a slightly lower concentration.

Nevertheless, we agree that there is a difference in the ion concentrations used in the simulations compared with the experimental work, and have therefore: (1) Toned down our previous statement to now read '*Nevertheless, our basic model of conductance through the monomeric unit of Ssp4 is supported by calculated and experimental conductance values falling within a similar range.*' (2) Added several sentences to acknowledge and discuss the difference in conditions (lines 390-394).

2. Inaccurate description of calcium permeability experiments.

The authors state that when CaCl₂ is in the trans chamber, "rectification was apparent at voltages > 50 mV" (Manuscript p. 7, lines 183-185). However, a close analysis of the provided raw data reveals this description to be inaccurate (Raw Data; Manuscript Fig. 4D). Rectification implies a voltage-dependent change in conductance where current continues to increase with voltage. The data, however, show that the current plateaus and ceases to increase at positive potentials. For instance, in one replicate, the current remains completely flat at 0.763 pA for all voltages from +50 mV to +80 mV. This phenomenon is more accurately described as current saturation.

This distinction is mechanistically important. Saturation suggests a rate-limiting step in permeation.

Recommendation: The authors should revise their description of the data in Figure 4C,D from "rectification" to the more accurate and mechanistically informative term "saturation."

We thank the Reviewer for this suggestion. We have now altered the wording within the manuscript accordingly (lines 182-184).

3. Potential confounding variables in calcium experiments remain unaddressed.

While the authors have correctly recalculated the Ca²⁺/K⁺ permeability ratio, they have dismissed the valid concern about the well-documented effects of Ca²⁺ ions on the physical properties of the phosphatidylethanolamine (PE) membranes used in their experiments. Their justification that this is "not intended to be a biophysical characterisation of the pore" and is "outside the scope of the current study" is insufficient when they proceed to interpret their data as evidence for a direct "Ca²⁺-dependent conformational change" within the Ssp4 protein itself (Manuscript p. 7, lines 189-190). Even with the more precise interpretation of current saturation, which points strongly to a protein-mediated effect, the authors cannot attribute the overall change in conductance solely to a protein conformational change without ruling out membrane-level effects.

Recommendation: The interpretation of a direct conformational change in the protein should be significantly toned down, and the potential for membrane-level effects should be explicitly discussed as a confounding variable.

We note that the wording in the place mentioned in the Results section is as follows: '*suggesting that the channel may undergo a Ca²⁺-dependent conformational change.*' We believe that this already indicates that the idea of direct conformational change in the protein is not definitive or necessarily the only interpretation or factor, and that it is reasonable to state, particularly given that the Reviewer also notes that what we observed 'points strongly to a protein-mediated effect'.

Nevertheless, we agree with the Reviewer that the surface charge of the lipid bilayer will influence the local ion concentration and electrostatic environment, which could modify the gating of the Ssp4 pore. However, it is widely reported that Ca²⁺ does not interact strongly with neutral PE lipids directly, unlike charged PS lipids. The neutral head group of PE lacks the strong electrostatic attraction and binding sites that Ca²⁺ ions require to alter channel function. It has been demonstrated, however, that Ca²⁺ binds to phosphate groups of all phospholipids, independent of their charge (doi: 10.1016/S0006-3495(00)76839-1) and this could have local effects at the membrane level. We have now explicitly discussed this in the manuscript (Discussion section, lines 423-427).

Conclusion

The authors have been exceptionally responsive to the initial reviews, and their revised manuscript presents a novel and exciting model for the function of the Ssp4 pore-forming toxin. The work is of high quality and potential impact. My recommendation is for acceptance following minor revisions that address the essential points of biophysical interpretation detailed above. Ensuring the manuscript's claims are supported by sound physical reasoning will solidify its conclusions and maximize its impact on the field.

We thank the Reviewer again for their positive comments and for their additional suggestions to improve the final manuscript.

Referee #3:

The authors addressed all of my concerns. I congratulate them on this elegant work.

We thank the reviewer for their comments and for taking the time to examine the revised manuscript.

Dear Sarah,

Thank you for addressing the final editorial requests. I am now happy to inform you that your manuscript has been accepted for publication in the EMBO Journal. Congratulations!

Meanwhile, I have written a short blurb that will accompany the title of your manuscript in our online table of contents. Please take a look and let me know if any corrections or adjustments are needed.

Blurb:

Serratia marcescens toxin Ssp4 disrupts membrane potential in recipient cells by forming ion-conducting pores.

If you have any questions, please do not hesitate to contact the Editorial Office. Thank you for your contribution to The EMBO Journal, and congratulations with a nice study!

With best wishes,

Ieva
